# Egr1 is a sex-dependent regulator of neuronal chromatin, structural plasticity, and behaviour

Devin Rocks [1], Luisa Demarchi [1], Laila Ouldibbat[1], Eric Purisic[1], Heining Cham[2], Eduardo F. Gallo [1], John M. Greally [3], Masako Suzuki [3,4] & Marija Kundakovic [1] ✉

Sex differences are found in brain structure and function across species, and across brain disorders in humans. A major source of brain sex differences is differential secretion of steroid hormones from the gonads across the lifespan. Specifically, ovarian hormones oestrogens and progesterone are known to dynamically change structure and function of the adult female brain, having a major impact on psychiatric risk. However, due to limited molecular studies in female rodents, very little is still known about molecular drivers of female-specific brain and behavioural plasticity. Here we show that ventral hippo-campal (vHIP) overexpression of Egr1, an oestrous cycle-dependent tran-scription factor, induces sex-dependent changes in vHIP neuronal chromatin, gene expression, and structural plasticity, along with female-specific effects on vHIP-dependent behaviours. Importantly, Egr1 overexpression and knock-down partially mimic the vHIP chromatin state associated with the high and low-oestrogenic phase of the oestrous cycle, respectively. We demonstrate that Egr1 directs neuronal chromatin opening across the sexes, although with limited genomic overlap. Our study not only reveals a sex-dependent chro-matin regulator in the brain, but also provides functional evidence that this sex-dependent gene regulation drives structural and behavioural plasticity, informing sex-based treatments for brain disorders.

Sex differences are found in brain structure and function across species, including mice and humans, and have been linked to reproductive and non-reproductive behaviours[1]. Studies have also reported a sig-nificant gender and sex bias in symptoms and prevalence across brain disorders in humans[2,3]. However, a better understanding of the sex-related factors and mechanisms shaping the healthy and diseased brain is needed to inform novel, sex-based treatments.

Sex-dependent characteristics of the brain are dynamically shaped throughout life, as a result of the actions of gonadal hormones, sex chromosomes, and the environment[1,4]. Brain sexual differentiation

is initiated by prenatal testosterone exposure in males, but gonadal hormones continue to play a major role in shaping the brain across the lifespan. Specifically, during the reproductive period in females, cycling ovarian hormones, oestrogens and progesterone, play a critical role in brain plasticity relevant to various behaviours including mating, feeding[5,6], learning[7,8], reward processing[9,10], and anxiety- and depression-related behaviours[11,12]. In addition, ovarian hormone shifts are associated with increased risk for neuropsychiatric conditions such as anxiety and depression disorders in women[11,13]. Thus, revealing molecular mechanisms and drivers underlying female-specific brain

[1]Department of Biological Sciences, Fordham University, Bronx, NY, USA. [2]Department of Psychology, Fordham University, Bronx, NY, USA. [3]Center for Epigenomics, Department of Genetics, Albert Einstein College of Medicine, Bronx, NY, USA. [4]Department of Nutrition, Texas A&M University, College Station, TX, USA. ✉e-mail: mkundakovic@fordham.edu

plasticity is of crucial importance for understanding sex-biased brain physiology and disease risk.

Historically, molecular studies in neuroscience have focused on the male brain[2,12,14,15], and our knowledge of sex-specific molecular mechanisms in the brain is therefore limited. It has been assumed that a specific group of transcription factors, known as immediate early gene (IEG) products, drive chromatin[16] and synaptic[17] plasticity across sexes, yet the molecular basis of sex differences in synaptic and behavioural plasticity remains obscure. Here, we leveraged previous findings from a physiological mouse model, which indicated that Egr1 is an oestrous cycle-dependent IEG and a candidate chromatin regulator in neurons of the ventral hippocampus (vHIP), driving synaptic and behavioural changes across the oestrous cycle in females[18] (Fig. 1a). We focused on the vHIP because of its critical role in emotion regulation and anxiety-related behaviour in mice[19].

Specifically, during the high-oestrogenic, proestrus phase of the cycle, we found increased vHIP chromatin accessibility surrounding Egr1 binding sites (Fig. 1a), coupled with increased Egr1 gene expression, increased vHIP dendritic spine density, and reduced anxiety indices, compared to the low-oestrogenic, dioestrus stage[18]. These findings suggested a model in which rising oestradiol induces neuronal Egr1 expression in the vHIP, after which Egr1 directs chromatin accessibility and transcription to drive oestrous cycle-dependent synaptic and behavioural changes. However, functional evidence validating this model is currently lacking.

Here we perform overexpression and knockdown of Egr1 in vHIP neurons in both sexes and provide functional evidence that Egr1 targets neuronal chromatin sex-dependently, with consequences for gene expression, structural plasticity, and anxiety- and depression-related behaviour. Since Egr1 manipulation mimics changes associated with physiological changes in ovarian hormones, our findings provide the foundation for sex-based treatments for female-biased disorders, strongly influenced by ovarian hormone shifts[11], such as anxiety and depression disorders.

## Results

### Egr1 overexpression in vHIP neurons induces proestrus-like behaviour in female mice

To better understand the physiological, cycle-dependent functions of Egr1, here we first determined *Egr1* expression in the vHIP across all phases of the mouse oestrous cycle (Fig. 1a-b). The rodent oestrous cycle is 4-5 days long, comprising four stages (proestrus, oestrus, metestrus, dioestrus), with cyclical oestrogen and progesterone levels (Fig. 1a). While *Egr1* shows a complex cyclical expression pattern across the oestrous cycle (Fig. 1b, Supplementary Data 1), its initial rise coincides with the rise in oestrogen in proestrus[18], increased expression of its target gene *Ppp1r1b* (Supplementary Figure 1, Supplementary Data 1) and reduced anxiety-related behaviour, as evidenced by increased time spent in the open arms of the elevated plus maze (EPM, Fig. 1c, Supplementary Data 1).

We further wanted to provide evidence that oestradiol levels directly regulate vHIP *Egr1* expression and behaviour across the oestrous cycle (Fig. 1d–f). To this end, we used "cyclical" treatment with oestradiol benzoate (EB; 1 μg s.c. every 4 days) in ovarian hormone-depleted, ovariectomized (OVX) mice for 6 weeks (Fig. 1d) mimicking the physiological rise in oestradiol occurring every 4 days in cycling females[20] (Fig. 1a). While removal of ovaries in young adult mice (at 8 weeks) led to reduced vHIP Egr1 mRNA levels (Fig. 1e) and increased anxiety-related behaviour in the open field (Fig. 1f) compared to age-matched, high-estrogenic (proestrus) cycling mice, the cyclical oestrogen replacement restored proestrus-like levels of *Egr1* expression (Fig. 1e, Supplementary Data 1) and partially rescued anxiety-related behaviour (Fig. 1f, Supplementary Data 1) in OVX mice. These results confirm the role of cyclical oestradiol in regulating vHIP Egr1 levels and behaviour. Together, these findings imply that oestradiol-induced *Egr1*

expression and Egr1's downstream transcriptional activity may drive behavioural changes across the cycle.

To determine whether increasing vHIP Egr1 expression by itself is sufficient to reproduce the proestrus-like behavioural state in the absence of ovarian hormones, we overexpressed Egr1 in vHIP neurons of OVX females (Fig. 1g, Supplementary Fig. 2). In this case, ovariectomy was performed pre-puberty to ensure that females used in these experiments never experienced ovarian hormone cycling (see *Methods*). These OVX females also show higher anxiety indices compared to high-oestrogenic proestrus females (Supplementary Fig. 3, Supplementary Data 1). Importantly, OVX females lack both fluctuations in ovarian hormones and in vHIP Egr1 levels otherwise present in intact females (Fig. 1a, b), thereby allowing us to isolate the independent effects of this previously identified sex-specific factor. While female gonadectomy precluded a direct comparison between the sexes (see *Methods*), we included age-matched males for a comparison in all experiments. Strikingly, OVX female mice overexpressing Egr1 in vHIP neurons exhibited reduced anxiety- and depression-related behaviours including increased centre entries in the open field, increased time in the open arms of the EPM, and reduced time spent immobile in the forced swim test, relative to eGFP-overexpressing controls (eGFP, Fig. 1h, Supplementary Data 1). By contrast, none of these behaviours were affected in males following Egr1 overexpression (Fig. 1i, Supplementary Data 1). Further, Egr1 overexpression did not affect overall locomotor activity in either sex (Supplementary Fig. 4, Supplementary Data 1), indicating that its effect is specific to anxiety- and depression-related behavioural indices, inducing proestrus-like behavioural state[18] in OVX females with no effect in intact males.

### Egr1 drives neuronal gene expression in a sex-dependent manner

With Egr1's well-defined role as an IEG and a transcription factor[21], we reasoned its sex-biased effects on behaviour could arise due to its influence on vHIP neuronal gene expression. vHIP neurons were of special interest because they are known to drive behaviours such as approach-avoidance in the EPM[19]. Thus, we performed the same AAV-mediated Egr1 overexpression experiment in male and OVX female mice, followed by gene expression analysis (RNA-seq) on purified neuronal (NeuN+) nuclei isolated from the targeted vHIP (Fig. 2a). Egr1 overexpression markedly altered neuronal gene expression in both sexes, with 1209 differentially expressed genes (DEGs) found in females and 1680 DEGs in males ($P_{adj} < 0.05$, Fig. 2b, Supplementary Data 2). As expected, *Egr1* was one such DEG and exhibited an AAV-mediated ~13-fold increase in expression in both females and males (Supplementary Fig. 5a, Supplementary Data 2). Whereas ovariectomy, again, prohibited a direct comparison between the sexes (see *Methods*), we performed overlaps of Egr1's transcriptional effects in males with those in OVX females to further isolate female-biased transcriptional mechanisms potentially underlying Egr1's behavioural effects. This analysis revealed that the majority of DEGs were sex-dependent, with only 37.3% (785/2104) of DEGs being shared between the sexes (Fig. 2c). Gene set enrichment analysis (GSEA) revealed male-biased effects on the expression of genes related to neuropeptide signalling and neurotransmitter transport (Fig. 2d). By contrast, female-biased enrichment includes genes related to excitatory synapses, thyroid hormone signalling, and, importantly, behavioural fear response (Fig. 2d, Supplementary Data 3), consistent with the behavioural effect we observed in females but not males (Fig. 1h, i).

Next, we performed a Gene Ontology (GO) enrichment analysis focused on the 424 DEGs identified exclusively in females (Fig. 2c), and found that these genes were enriched for numerous terms related to synaptic function (Fig. 2e), which implicates Egr1 in sex-dependent regulation of a set of synapse-related genes in females. While effects of Egr1 overexpression likely result from its cumulative impact on gene expression, we identified several anxiety-relevant example genes with

 

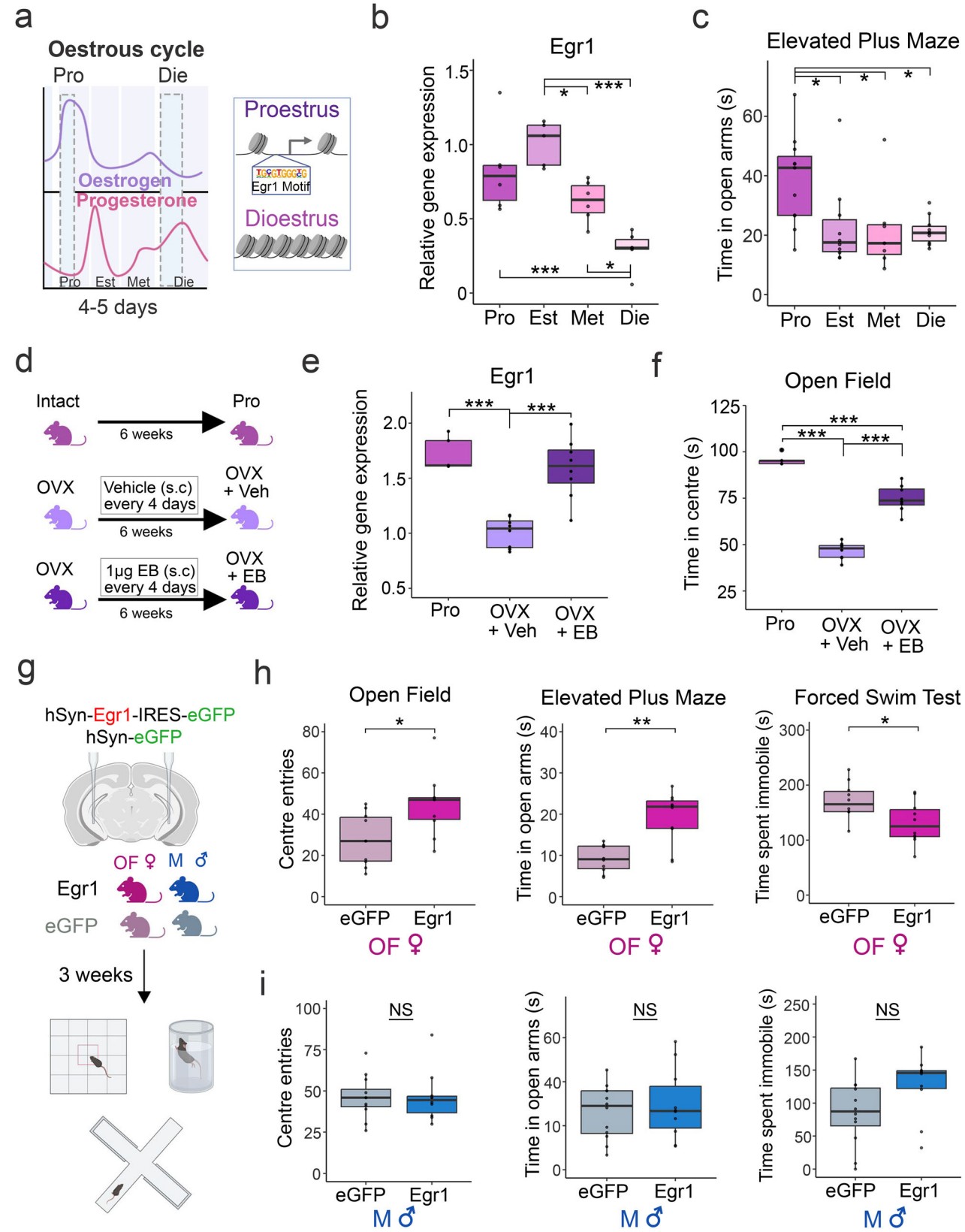

expression patterns that are concordant with Egr1's behavioural effects (Fig. 2f). *Atp1a2*, for instance, is a gene in the Behavioral Fear Response term enriched specifically in females (Fig. 2d). *Atp1a2* deficient-mice exhibit enhanced fear and anxiety behaviours[22], and we found that *Atp1a2* expression in vHIP neurons is upregulated in females and unchanged in males following Egr1 overexpression (Fig. 2f). Another

example is *Ppp3r1*, encoding calcineurin B, which is a member of multiple synapse-related terms that were enriched in female-specific DEGs (Fig. 2e). Systemic inhibition of calcineurin was found to increase anxiety- and depression-related behaviours in male mice[23], and its expression is upregulated in females and unchanged in males after Egr1 overexpression (Fig. 2f). Interestingly, for both of those genes,

**Fig. 1 | Neuronal Egr1 expression drives sex-specific behaviours. a** An illustration depicts changes in oestrogen and progesterone levels over a typical 4-5 day oestrous cycle in mice, with considerable reorganization of chromatin enriched for Egr1 binding sites between proestrus and dioestrus phases. Here we show cyclical *Egr1* expression in the ventral hippocampus (*n* = 5 Est, 6 Pro/Met/Die; normalized to *Ppia* endogenous reference gene expression) (**b**) and elevated plus maze behaviour (*n* = 11 Pro, 10 Est/Die, 7 Met) (**c**) across the oestrous cycle in female mice. We also observe that, utilizing a cyclical oestradiol dosing protocol (**d**), ovariectomy-induced deficits in both ventral hippocampal *Egr1* expression (**e**, *n* = 5 Pro, 8 OVX +Veh/OVX + EB) and time spent in the centre of the open field (**f**, *n* = 5 Pro, 8 OVX +Veh/OVX + EB) can be fully or partially rescued, respectively. AAV-mediated Egr1 overexpression in ventral hippocampal neurons in vivo affects a battery of anxiety- and depression-related behaviours in a sex-specific manner (**g**), by mimicking the high-oestrogenic proestrus phase in ovariectomized female (OF) mice (**h**, *n* = 10/

group) with no effect in males (M) (**i**, *n* = 10 Egr1, 12 eGFP), as compared to their respective eGFP controls. Data in (**b**, **c**, **e**, **f**) were analysed with one-way ANOVA with Holm's post hoc test. Data in panels h and i were analysed with a Welch two-sample t-test (two-sided). Box plots (box, 1st-3rd quartile; horizontal line, median; whiskers, 1.5xIQR); *$p < 0.05$; **$p < 0.01$; ***$p < 0.001$; NS non-significant (exact *p*-values provided in Supplementary Data 1). Pro proestrus (purple), Est oestrus (light-purple), Met metestrus (pink), Die dioestrus (light-pink), OVX ovariectomized at 8 weeks, OVX+Veh vehicle-treated OVX group (light-violet), OVX + EB oestradiol benzoate-treated OVX group (dark-violet), eGFP OF eGFP-injected females ovariectomized at 4 weeks (pale-pink), Egr1 OF Egr1-injected females ovariectomized at 4 weeks (bright-pink), eGFP M eGFP-injected males (pale-blue), Egr1 M Egr1-injected males (bright-blue). Source data are provided as a Source Data file. Schematics in (**a**, **d**, **g**) were created in BioRender. Rocks, D. (2025) https://BioRender.com/oo9h37f.

males seem to show a higher basal level of expression compared to OVX females, while this sex bias is reduced following Egr1 over-expression. Finally, *Npsr1*, encoding the neuropeptide S receptor 1, was the only gene that exhibited the opposite pattern of expression changes across sex after Egr1 overexpression (Supplementary Fig. 5b). *Npsr1* has been implicated in altered anxiety indices in mice[24] and humans[25], and, notably, its expression is upregulated in females and downregulated in males (Fig. 2f, Supplementary Fig. 5b) following Egr1 overexpression.

### Egr1 opens neuronal chromatin in a sex-dependent manner

Given our previous work implicating Egr1 in chromatin regulation within vHIP neurons over the oestrous cycle in females[18,26], we wondered whether altered chromatin organization is a feature of Egr1's sex-dependent gene regulatory functions. To this end, following vHIP-targeted Egr1 overexpression, we performed chromatin accessibility (ATAC-seq) analysis on vHIP neuronal (NeuN + ) nuclei purified from male and OVX female mice (Fig. 3a). Across all samples, we identified a total of 222,253 accessible chromatin regions in vHIP neurons. Of these, 16.0% (35,491) in females and 14.0% (31,105) in males were identified as differentially accessible regions (DARs; p_adj < 0.05), indicating that Egr1 overexpression led to substantial reorganization of neuronal chromatin (Fig. 3b, Supplementary Data 4). Importantly, the vast majority of DARs in both females (30,565 or 86.1%) and males (26,546 or 85.3%) *gained* accessibility following Egr1 overexpression. This finding is in line with our hypothesis that Egr1 is an "opener" of chromatin[18], and is consistent with previous findings of another IEG, cFos, inducing chromatin accessibility following neuronal activity[16]. Thus, we focused our subsequent analyses on the regions that gained accessibility following Egr1 overexpression (Egr1 gained-open regions). Following the same approach we used for RNA-seq data, we performed an overlap of Egr1 gained-open regions in OVX females and males. This analysis revealed that only 56.4% (20,612/36,499) of gained-open regions are shared between females and males, while 27.3% (9953/36,499) were uniquely identified in females and 16.3% (5934/36,499) were uniquely identified in males (Fig. 3b).

We next asked whether these regions gain accessibility indirectly, downstream of Egr1's transcriptional effects, or alternatively, if they may be targeted by Egr1 specifically, as a function of their DNA sequence. Strikingly, motif analysis revealed, in both sexes, that more than 70% of regions that gain accessibility after Egr1 overexpression contain the Egr1 motif (Fig. 3c, e.g., Supplementary Fig. 6, Supplementary Data 5), providing evidence that Egr1 mediates opening of neuronal chromatin targeted to specific motif-containing regions. While this was previously demonstrated for cFos in the male brain[16], our study reveals an IEG product that acts as a sex-dependent molecular driver of chromatin accessibility in neurons.

To explore the functional role of Egr1-induced gained-open regions containing Egr1 binding sites, we overlapped their sequences

with previously published ChIP-seq data on histone modifications in hippocampal neurons[27]. Importantly, we found relative enrichment of H3K4me1 and H3K27ac, marking active enhancers[28], and depletion of H3K4me3, marking gene promoters[29] (Fig. 3d), indicating that neuronal enhancers are the primary substrate for targeted opening of chromatin by Egr1. Similar to the total Egr1-induced gained-open regions, an overlap between males and females of gained-open regions with Egr1 binding sites revealed that 46.2% (12,911/27,932) are unique to one sex (Fig. 3e). Pathway analysis on genes annotated to these sex-dependent regions show male-biased enrichment for pathways related to BDNF signalling (Fig. 3e), in line with male-biased expression changes related to Neuropeptide Signaling (Fig. 2d), and female-biased enrichment for pathways related to synapse regulation and, importantly, anxiety-related behaviour (Fig. 3e), consistent with the female transcriptional enrichment for Behavioral Fear Response genes (Fig. 2d) and the effects of Egr1 on anxiety-related behaviour in females (Fig. 1h). Together, this indicates that Egr1 initiates chromatin opening near different gene sets with distinct functions in males and females.

### Egr1-induced chromatin opening mediates gene expression changes

By overlapping ATAC-seq and RNA-seq data from the Egr1 over-expression experiments, we next explored whether Egr1's chromatin effects were associated with effects on gene expression. A general overlap revealed that the majority of DEGs in both sexes (Fig. 2b), 62.7% (758/1209) in females and 61.6% (1035/1680) in males, are associated with at least one chromatin region that changes accessibility following Egr1 overexpression (Supplementary Fig. 7). This indicated that, in general, Egr1-induced chromatin changes mediated gene expression changes. However, we wondered which subset of these Egr1-induced DEGs were associated specifically with opening of chromatin targeted to the Egr1 motif. In females, 51.2% (619/1209) of DEGs are associated with gained-open chromatin containing the Egr1 motif (Fig. 3f). Of these DEGs, 40.7% (252/619) are female-specific and enriched for terms and pathways related to synaptic function and cell adhesion (Fig. 3f). An illustrative example demonstrating Egr1's sex-dependent effects on gene regulation is *Nrn1* (Fig. 3g), a neurotrophic factor that has been shown to regulate hippocampal dendritic spine density[30]. Egr1 overexpression induces chromatin opening within the Egr1 motif-containing region downstream of the *Nrn1* locus in females only, which is further associated with a female-specific increase in gene expression (Fig. 3g).

In males, we observed a similar involvement of Egr1's targeted chromatin opening in gene regulation, where 52.1% of all DEGs (875/1680) are associated with Egr1 motif-containing gained-open chromatin (Fig. 3h). Of these, 58.1% (508/875) are male-specific DEGs and are enriched for terms and pathways related to synapses, cation channels, and neurotransmitter receptors (Fig. 3h). An example is *Grin2a*, encoding a subunit of the NMDA glutamate receptor which

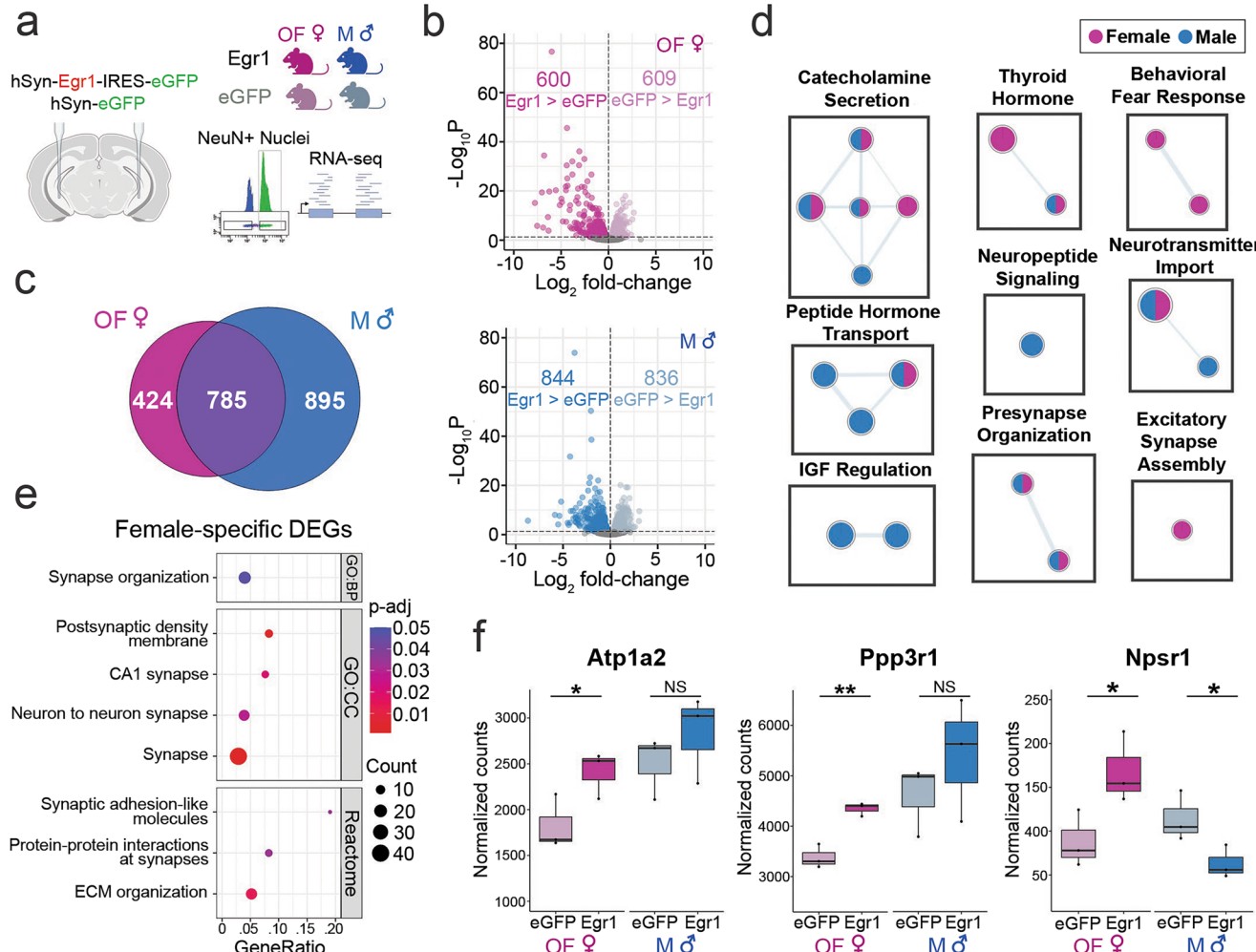

**Fig. 2 | Egr1 overexpression induces sex-dependent gene expression changes in vHIP neurons. a** RNA-sequencing was performed on neuronal (NeuN + ) nuclei purified from the vHIP of males (M) and ovariectomized females (OF) injected with either Egr1 or eGFP AAVs carrying the neuron-specific hSyn promoter. **b** >1000 genes show changes in gene expression in females and males following Egr1 overexpression (P$_{adj}$ < 0.05, Benjamini-Hochberg correction for multiple testing). **c** Only 37.3% (785/2104) of DEGs are shared between the sexes. **d** GSEA analysis shows sex-specific terms including Behavioral Fear Response; Enrichment Map depicts nodes representing terms/pathways enriched in one or more gene list (denoted by node colour), while edges represent genes shared between pathways. **e** Female-specific DEGs are enriched for synaptic function-related GO terms; Dot plot colours indicate *p*-values, and dot size corresponds to the number of overlapping genes between the list and the term/pathway (Count); the x-axis corresponds to the ratio of genes in the input list to genes in the term/pathway

(GeneRatio). **f** Sex-specific DEGs include genes implicated in anxiety- and depression-related behaviours (*Atp1a2*, *Ppp3r1*, *Npsr1*). RNA-seq data for individual genes are shown using normalized count plots (*n* = 3 biological replicates/group/ sex). RNA-seq data were analysed with the Wald statistical test in DESeq2 (two-sided). Enrichment analysis was performed using hypergeometric tests. Box plots (box, 1st-3rd quartile; horizontal line, median; whiskers, 1.5xIQR); *p$_{adj}$ < 0.05; **p$_{adj}$ < 0.01; NS non-significant, Benjamini-Hochberg correction for multiple testing (exact *p*-values provided in Supplementary Data 2). GO Gene Ontology, BP Biological Process, CC Cellular Component, eGFP OF eGFP-injected ovariectomized females (pale-pink), Egr1 OF Egr1-injected ovariectomized females (bright-pink), eGFP M eGFP-injected males (pale-blue); Egr1 M Egr1-injected males (bright-blue). Schematic in panel a was created in BioRender. Rocks, D. (2025) https://BioRender.com/oo9h37f.

---

plays an important role in synaptic plasticity[31]. An Egr1 motif-containing intronic region becomes more accessible after Egr1 overexpression in males but not females, which corresponds to male-specific upregulation of *Grin2a* expression (Fig. 3i). These results demonstrate that Egr1 overexpression mediates opening of neuronal chromatin surrounding Egr1 binding sites in both sexes, with distinct genomic regions targeted in males and females associated with sex-dependent effects on gene expression.

**Egr1 overexpression partially recapitulates the proestrus-associated transcriptional program in vHIP neurons**

Having demonstrated that Egr1 overexpression in vHIP neurons reproduces a proestrus-like behavioural phenotype in OVX females (Fig. 1h), we next examined whether Egr1 overexpression similarly

recapitulates molecular phenotypes associated with physiological ovarian hormone cycling in vHIP neurons. Leveraging our previously generated RNA-seq data, which measured gene expression in vHIP neurons over the oestrous cycle[18], we overlapped the identified cycle-related DEGs with DEGs resulting from Egr1 overexpression in OVX females. This analysis revealed that 18.6% (32/172) of oestrous cycle DEGs similarly undergo expression changes in OVX females overexpressing Egr1 (Fig. 4a, Supplementary Data 2c). As expected, this is a small fraction of total DEGs induced by viral overexpression of Egr1 (32/3023 or 1.1%), yet it is consistent with the 24.1% of oestrous cycle DEGs with chromatin changes that we previously found to have Egr1 binding motifs[18]. Two example genes which illustrate the role of Egr1-driven gene expression in oestrous cycle-dependent behavioural phenotypes are *Arid1a* and *Gatad2b*. *Arid1a* encodes a subunit of the

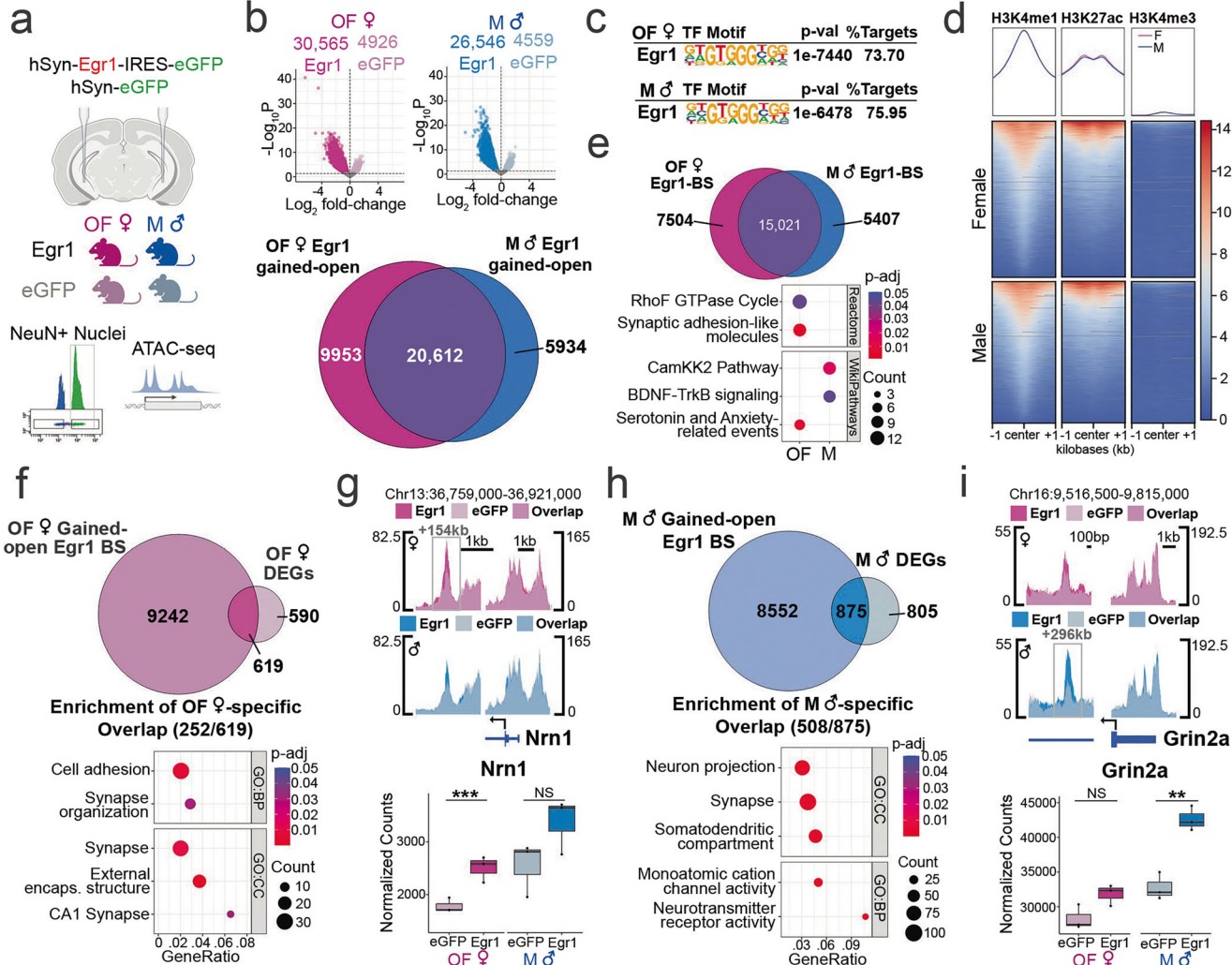

**Fig. 3 | Egr1 overexpression induces sex-dependent chromatin changes in vHIP neurons. a** ATAC-seq was performed on neuronal (NeuN + ) nuclei purified from the vHIP of males (M) and ovariectomized females (OF) injected with either Egr1 or eGFP AAVs carrying the neuron-specific hSyn promoter. **b** Egr1 overexpression induces extensive re-organization (primarily opening) of vHIP neuronal chromatin, with only 56.4% (20,612/36,499) of gained-open regions being shared across sexes. **c** In both males and females, Egr1-induced open regions are overwhelmingly (>70%) enriched for the Egr1 motif; Egr1 motif-containing gained-open regions: (**d**) are primarily located in enhancers (overlapping with the enrichment of H3K4me1/ H3K27ac but not H3K4me3 histone marks) and (**e**) affect different sets of genes in males and females. **f** In females, 51.2% (619/1209) of DEGs are associated with gained-open chromatin containing the Egr1 motif, and, of those, female-specific DEGs are enriched for synaptic function-related genes. **g** An exemplary gene is *Nrn1* which is associated with female-specific chromatin opening and gene expression. **h** In males, 52.1% of all DEGs (875/1680) are associated with Egr1 motif-containing gained-open chromatin and, of those, male-specific DEGs are enriched for a different set of synaptic function-related genes. **i** An exemplary gene is *Grin2a* which is associated with male-specific chromatin opening and gene expression. Dot plots;

colours indicate p-values and dot size corresponds to the number of overlapping genes between the list and the term/pathway (Count); the x-axis corresponds to the ratio of genes in the input list to genes in the term/pathway (GeneRatio). ATAC-seq data is shown using SparK plots of group-average normalized ATAC-seq reads ($n = 4$ biological replicates/group/sex). RNA-seq data is shown using normalized count plots ($n = 3$ biological replicates/group/sex). Both RNA-seq and ATAC-seq data were analysed using the Wald statistical test in DESeq2 (two-sided). Enrichment analysis and motif analysis were performed using hypergeometric tests. Box plots (box, 1st-3rd quartile; horizontal line, median; whiskers, 1.5xIQR); **$p_{adj}$ < 0.01; ***$p_{adj}$ < 0.001, NS non-significant, Benjamini-Hochberg correction for multiple testing (exact p-values for RNA-seq, ATAC-seq, and motif analysis are provided in Supplementary Data 2, Supplementary Data 4, and Supplementary Data 5, respectively). GO Gene Ontology, BP Biological Process, CC Cellular Component, eGFP OF eGFP-injected ovariectomized females (pale-pink), Egr1 OF Egr1-injected ovariectomized females (bright-pink), eGFP M eGFP-injected males (pale-blue), Egr1 M Egr1-injected males (bright-blue). Schematic in panel a was created in BioRender. Rocks, D. (2025) https://BioRender.com/oo9h37f.

SWI/SNF chromatin remodelling complex, perturbations of which have been associated with anxiety-related phenotypes[32]. Importantly, the up-regulation of *Arid1a* expression observed during the high-estrogenic, proestrus phase in cycling females is mimicked by Egr1 overexpression in OVX females (Fig. 4b). *Gatad2b* encodes a subunit of the NuRD chromatin remodelling complex, whose levels are associated with anxiety-related behaviour[33], and is also concordantly upregulated in proestrus females and OVX females overexpressing Egr1 (Fig. 4b).

## Egr1 overexpression partially recapitulates proestrus-associated chromatin changes in vHIP neurons

We next performed overlaps of chromatin accessibility (ATAC-seq data) between the oestrous cycle[18] and Egr1 overexpression experiments. Interestingly, compared to gene expression, we see a much greater overlap between the ATAC-seq datasets, where 71.2% (6847/ 9614) of genes with differential chromatin peaks across the oestrous cycle also show changes in chromatin in OVX females after Egr1 overexpression (Supplementary Fig. 8). Since we specifically

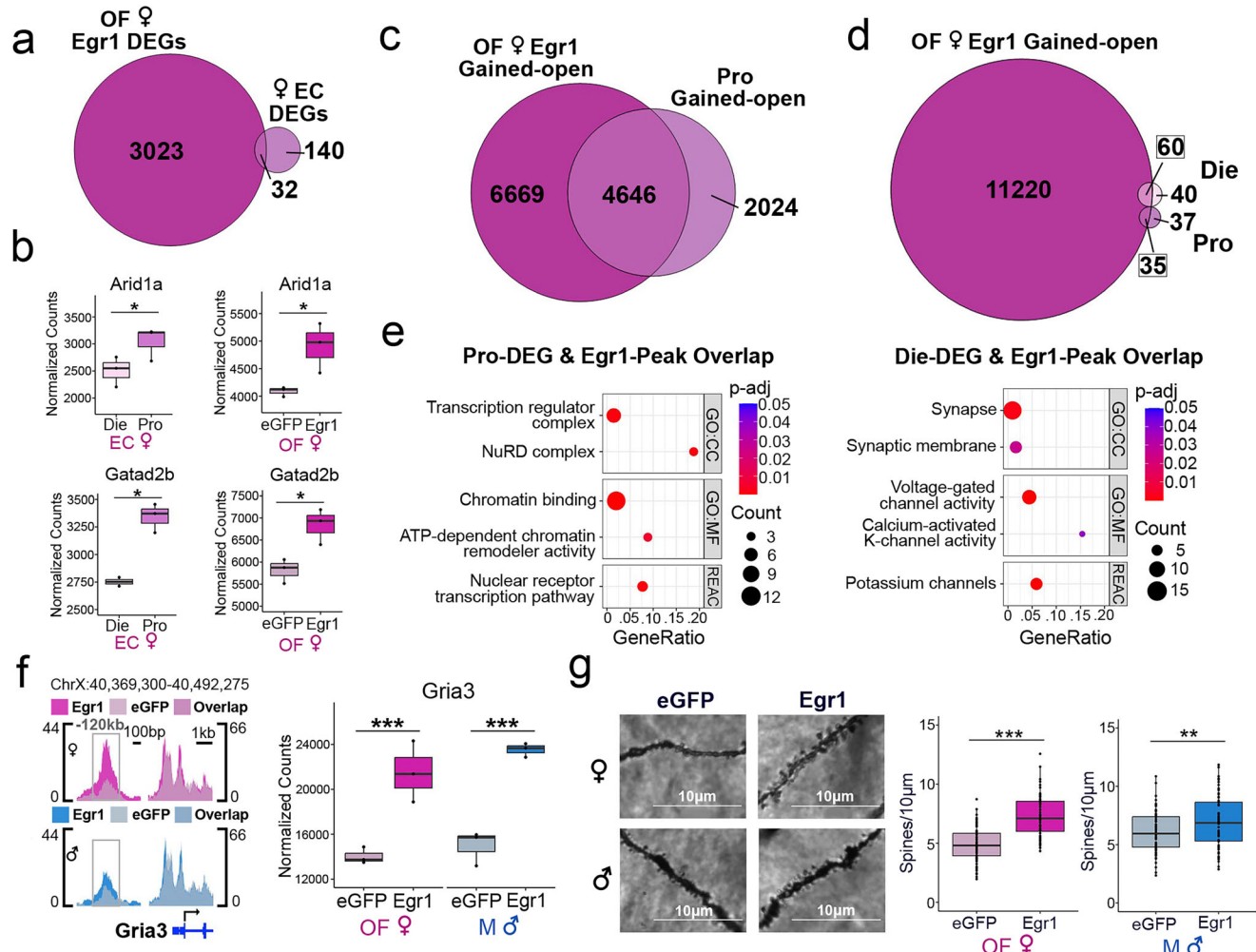

**Fig. 4 | Egr1 overexpression partially recapitulates proestrus-associated changes in gene expression, chromatin, and synaptic plasticity. a** 18.6% of oestrous cycle (EC)-related DEGs[18] exhibit expression changes following Egr1 overexpression in ovariectomized females (OF; $p_{nominal} < 0.05$ threshold used for overlap) including. **b** *Arid1a* and *Gatad2b* that are upregulated in proestrus and induced by Egr1 overexpression (*$p < 0.05$; datasets were normalized and analysed separately; $n = 3$ biological replicates/group). **c** 69.7% of proestrus (Pro) gained-open chromatin regions similarly gain accessibility after Egr1 overexpression in females. **d** Oestrous cycle-, Pro- and dioestrus (Die)-, related DEGs exhibit an even higher overlap (55.2%) with genes bearing Egr1 gained-open regions. **e** The overlapping genes that are induced in Pro and those that are induced in Die are enriched for different terms and pathways. Dot plots colours indicate *p*-values, and dot size corresponds to the number of overlapping genes between the list and the term/pathway (Count); the x-axis corresponds to the ratio of genes in the input list to genes in the term/pathway (GeneRatio). **f** *Gria3* is a synaptic function-related gene that shows changes in chromatin accessibility and gene expression following Egr1 overexpression in both sexes (***$p_{adj} < 0.001$, Benjamini-Hochberg correction for multiple testing). ATAC-seq data is shown using SparK plots of group-average

normalized ATAC-seq reads ($n = 4$ biological replicates/group/sex). RNA-seq data is shown using count plots ($n = 3$ biological replicates/group/sex). **g** Egr1 overexpression increases dendritic spine density (data from $n = 5$ mice/group/sex; 16 dendrite segments counted per animal) in the CA1 vHIP neurons of both ovariectomized females (OF) and males (M), but the effect size is higher in females ($d = 1.532$, ***$p < 0.001$) compared to males ($d = 0.4468$, **$p < 0.01$). RNA-seq and ATAC-seq data were both analysed with the Wald statistical test in DESeq2 (two-sided), enrichment analysis was performed using hypergeometric tests, and dendritic spines data were analysed with Welch two-sample t-test (two-sided). Exact p-values for dendritic spines, RNA-seq, and ATAC-seq provided in Supplementary Data 1, Supplementary Data 2, and Supplementary Data 4, respectively. Box plots (box, 1st-3rd quartile; horizontal line, median; whiskers, 1.5xIQR), GO Gene Ontology, CC Cellular Component, MF Molecular Function, Reac Reactome, Die dioestrus (light-pink), Pro proestrus (purple), eGFP OF eGFP-injected ovariectomized females (pale-pink), Egr1 OF Egr1-injected ovariectomized females (bright-pink), eGFP M eGFP-injected males (pale-blue), Egr1 M Egr1-injected males (bright-blue). Source data are provided as a Source Data file.

implicated Egr1 in mediating the proestrus behavioural and transcriptional state[18], we next overlapped Egr1 gained-open regions with proestrus gained-open regions and found 69.7% (4646/6670) of genes with proestrus gained-open chromatin also exhibit gained-open chromatin after Egr1 overexpression (Fig. 4c). Due to the extensive overlapping effects of Egr1 overexpression and proestrus on chromatin accessibility, we wondered whether chromatin effects are more reflective of Egr1's physiological effects across the cycle than its effects on gene expression, which likely rely on the presence of additional transcription factors that may be absent in the vHIP neurons of OVX

females. We therefore explored the overlap between Egr1's chromatin effects and oestrous cycle DEGs, finding that 55.2% (95/172) of oestrous cycle DEGs have Egr1 gained-open chromatin after Egr1 overexpression in OVX females (Fig. 4d). This larger overlap was expected, in part, due to the large number of genes associated with gained accessibility after Egr1 overexpression (11,315). Importantly, however, we could now further explore the functional significance of genes that simultaneously exhibit Egr1-mediated chromatin regulation and oestrous cycle-mediated transcriptional changes. Of these oestrous cycle DEGs that overlap with Egr1-induced chromatin changes, 68.4% (65/95) also

have chromatin changes across the oestrous cycle, indicating the importance of chromatin accessibility in their physiological regulation (Supplementary Fig. 9). Further, consistent with Egr1's role in both activating[34] and repressing[35] transcription, we found Egr1 gained-open regions overlap with genes both up- and downregulated in proestrus compared to dioestrus females (Fig. 4d). Overlapping genes upregulated in proestrus (down in dioestrus) are enriched for terms and pathways related to chromatin and transcription regulation, while overlapping genes downregulated in proestrus (up in dioestrus) are enriched for terms related to synaptic function and ion channels (Fig. 4e). These results indicate that Egr1 mediates opening of chromatin near its binding motifs near genes that play a critical role in transcriptional regulation and neuronal function within physiologically cycling female mice.

## Egr1 overexpression recapitulates proestrus-associated changes in structural plasticity in vHIP neurons

Notably, we found enrichment of genes relevant to synaptic function among female-specific Egr1-induced DEGs (Fig. 2e) and in the overlap between Egr1's chromatin effects and oestrous cycle-dependent DEGs (Fig. 4e). Thus, we hypothesized that Egr1's gene regulatory effects may drive oestrous cycle-dependent changes in the density of dendritic spines, the sites of active synapses. As an example, *Gria3* is a putative Egr1 target gene that encodes a subunit of the AMPA glutamate receptor, which is implicated in the regulation of dendritic spines[36,37]. We previously found that vHIP *Gria3* exhibited increased expression and a more open chromatin state during the proestrus (compared to dioestrus) phase of the oestrous cycle[18]. Notably, Egr1 overexpression mimics proestrus in OVX females, leading to an increase in chromatin accessibility at a region upstream of *Gria3* (Fig. 4f), containing an Egr1 binding site, and increased *Gria3* gene expression (Fig. 4f). Interestingly, basal *Gria3* levels are similar between OVX females and males, and thus Egr1 overexpression leads to similar effects on *Gria3* chromatin and gene expression in both sexes (Fig. 4f).

Following up on this regulation of genes involved in synapse regulation, we next assessed whether Egr1 overexpression alters dendritic spine density in vHIP neurons. After delivering Egr1 or control AAVs to the vHIP of OVX females and males, we performed Golgi-Cox staining and imaged dendrites of ventral CA1 pyramidal neurons. We found that Egr1 overexpression increases vHIP dendritic spine density in OVX females (Fig. 4g, Supplementary Data 1), again mimicking the proestrus phase of the cycle[18]. Egr1 overexpression also increased spine density in males, albeit with a smaller effect size (d = 0.4468, compared to d = 1.532 in females; Supplementary Data 1).

## Egr1 knockdown reduces expression and chromatin accessibility of oestrogen response genes

While overexpression of Egr1 recapitulated a number of oestrous cycle-dependent phenotypes in OVX females, we wondered whether reducing Egr1 expression in intact *proestrus* females could induce the opposite effect and disrupt the proestrus-driven transcriptional program in vHIP neurons. To this end, we injected a neuron-specific AAV expressing either an Egr1-targeting shRNA or a scramble control into the vHIP of proestrus females and males. We then purified vHIP neuronal (NeuN + ) nuclei and performed RNA-seq and ATAC-seq (Fig. 5a).

Analysis of the RNA-seq data revealed that Egr1 knockdown led to 91 DEGs ($p_{adj} < 0.05$; Fig. 5b, Supplementary Data 6), virtually all of which were downregulated in the knockdown group (97.8%; 89/91). This reduced number of DEGs compared to Egr1 overexpression can be explained, in part, by the comparatively smaller effect of knockdown on Egr1 expression (~1.6 fold reduction, Fig. 5b; compared to ~13-fold increase via overexpression, Supplementary Fig. 5a). Importantly, however, the DEGs we did identify were enriched for oestrogen response genes (Fig. 5c, Supplementary Data 6c), including *Sh3bp5*,

which has been previously identified as part of a MAPK-related oestrogen-induced transcriptional program[38], and *Rara*, encoding the retinoic acid receptor alpha which has been shown to cooperate with oestrogen receptors to mediate oestrogen-induced transcriptional regulation[39] (Fig. 5d). Further, while proestrus females and males exhibit similar baseline levels of Egr1 expression (Fig. 5b), consistent with our previous findings[18], our use of intact females here allowed us to incorporate sex as a factor in our statistical model (see *Methods*) to directly investigate whether Egr1 knockdown exhibits any sex-biased effects on gene expression. We therefore performed GSEA on ranked gene lists from the analysis of group (i.e., shRNA vs. scramble in both sexes) and group-by-sex interaction effects on gene expression. This revealed that Egr1 knockdown altered expression of genes related to calcium signalling and synapses, with sex-biased effects on the expression of protein synthesis- and dendrite membrane-related gene sets (Fig. 5e, Supplementary Data 7).

Similar to the gene expression results, analysis of the ATAC-seq data revealed a modest number of DARs (301, $P_{adj} < 0.05$; Fig. 5f, Supplementary Data 8a). Egr1 knockdown exclusively reduced chromatin accessibility at these DARs (Fig. 5f), and genes annotated to these closed regions were again enriched for oestrogen response genes (Fig. 5g, Supplementary Data 8c). Remarkably, motif analysis of these closed regions revealed that nearly 80% of them contain the Egr1 motif (Fig. 5h, Supplementary Data 9). This result resembles motif enrichment we observed with chromatin opening following Egr1 overexpression (Fig. 3c) and implies that Egr1 expression is required for both the induction and maintenance of accessibility at Egr1 motifs. To more definitively link Egr1 genomic binding to its chromatin regulatory functions, we assessed Egr1 binding activity in regions of interest by leveraging previously published Egr1 ChIP-seq data from the male cortex[40]. For this analysis, we focused on regions of chromatin that are putatively regulated by Egr1 based on: i) gained accessibility in the overexpression experiment (Supplementary Fig. 10a); ii) lost accessibility in the knockdown experiment (Supplementary Fig. 10b); and iii) gained accessibility in proestrus compared to dioestrus[18] (Supplementary Fig. 10c). In all cases, we found that these regions are highly enriched for Egr1 binding in vivo (Supplementary Fig. 10). Further, we again found that Egr1 motif-containing regions were enriched for enhancer marks (Fig. 5i), consistent with our observations from the overexpression experiment (Fig. 3d), along with relatively higher enrichment for the promoter mark H3K4me3 (Fig. 5i).

Since the previous publication on Egr1 binding in the male cortex reported that ~44% of binding sites are at promoters[40], we wondered whether the observed enhancer enrichment here was due to a general inaccessibility of cortical Egr1-bound promoters in the ventral hippocampus, or whether they are simply not differential between groups. To answer this question, we overlapped Egr1-bound promoters identified in the cortical ChIP-seq analysis with either all unique ATAC-seq peaks (Supplementary Fig. 11a) or unique Egr1 vs. eGFP DARs (Supplementary Fig. 11b). This analysis demonstrated that while the majority of cortical promoters bound by Egr1 are accessible in vHIP neurons, very few are differentially accessible between the Egr1 and eGFP conditions. This is consistent with our observation that the gained accessibility surrounding Egr1 motifs that we observed following Egr1 overexpression is targeted to neuronal enhancers, rather than promoters.

Taken together, our Egr1 overexpression and knockdown results indicate that Egr1 targets and maintains the accessibility of neuronal enhancers, driving sex-dependent transcriptional programs related to neuronal function and, importantly, the neuronal response to oestrogen.

## Egr1 knockdown partially reverses the proestrus chromatin state in females

We next considered the extent to which Egr1 knockdown in proestrus females recapitulates a dioestrus-like chromatin state. In addition to the 301 group DARs we identified (Fig. 5f), we identified regions with

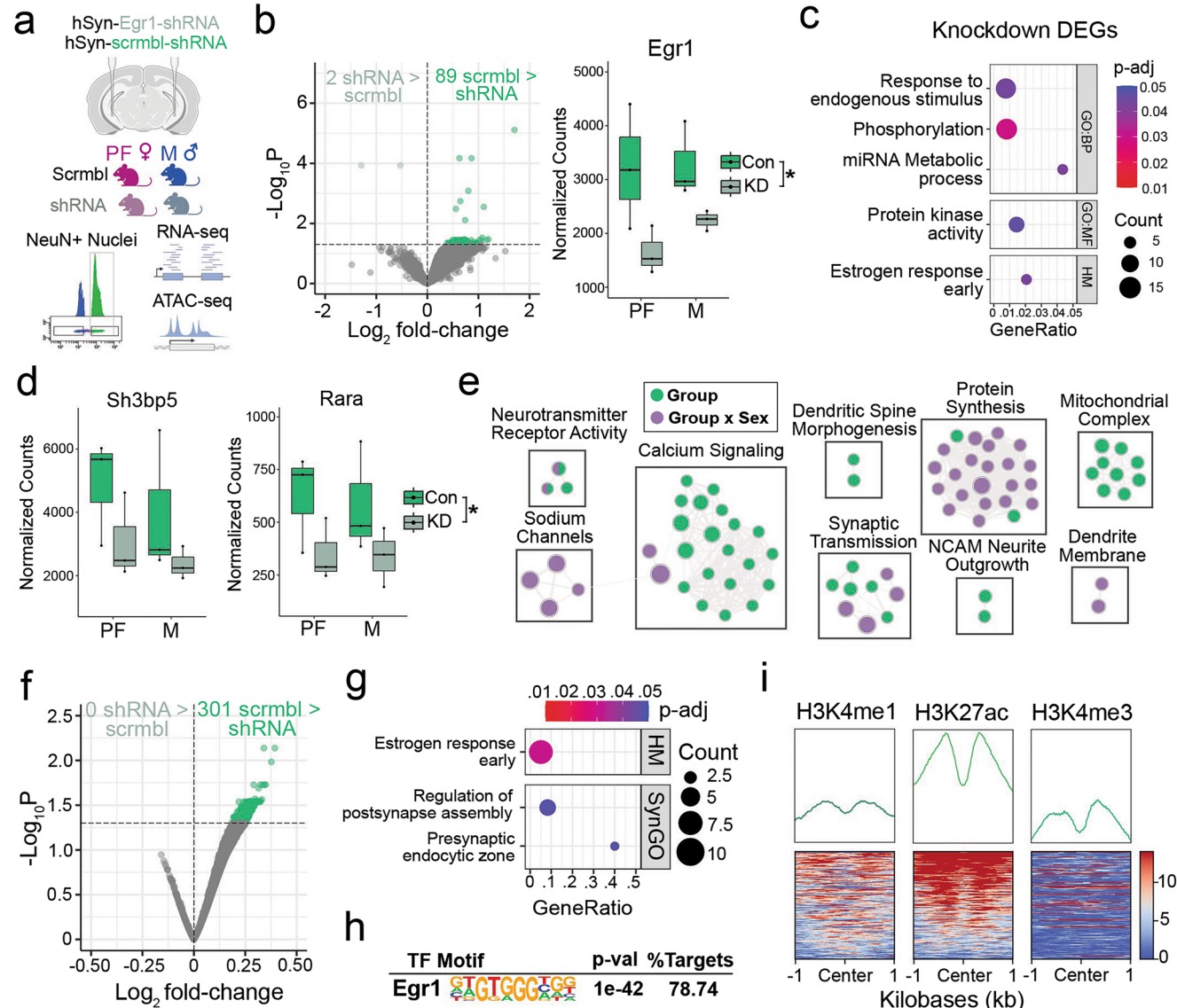

**Fig. 5 | Egr1 knockdown effects on transcription and chromatin are enriched for oestrogen response genes. a** RNA-seq and ATAC-seq were performed on neuronal (NeuN +) nuclei purified from the vHIP of males (M) and proestrus females (PF) injected with either Egr1-targeting or scrambled shRNA AAVs carrying the neuron-specific hSyn promoter. **b** Egr1 shRNA alters the expression of 91 genes (left; $p_{adj} < 0.05$; Benjamini-Hochberg correction for multiple testing) and results in a 1.6-fold reduction in neuronal Egr1 levels (right). **c** Differentially expressed genes (DEGs) are enriched for oestrogen response genes, which include **d** *Sh3bp5* (left) and *Rara* (right), both of which are downregulated following Egr1 knockdown (*, $p_{adj} < 0.05$). **e** GSEA reveals sex-dependent changes in expression of genes related to regulation of dendrites and protein synthesis following Egr1 knockdown; Enrichment Map depicts nodes representing terms/pathways enriched in one or more gene list (denoted by node colour; group analysis in green and group by sex interaction in purple), while edges represent genes shared between pathways. **f** 301 chromatin regions become less accessible following Egr1 knockdown ($p_{adj} < 0.05$; Benjamini-Hochberg correction for multiple testing), and (**g**) are enriched for oestrogen response genes and (**h**) Egr1 motifs. **i** Egr1 knockdown closes Egr1 motif-

containing chromatin regions enriched for the enhancer marks (H3K4me1 and H3K27ac), along with limited enrichment for the promoter mark H3K4me3. RNA-seq data is shown using count plots ($n = 3$ biological replicates/group/sex). RNA-seq and ATAC-seq data were both analysed with the Wald statistical test in DESeq2 (two-sided), while motif and enrichment analyses were performed using hypergeometric tests. Exact p-values for RNA-seq, and ATAC-seq, and motif analysis are provided in Supplementary Data 6, Supplementary Data 8, and Supplementary Data 9, respectively. Box plots (box, 1st-3rd quartile; horizontal line, median; whiskers, 1.5xIQR). Dot plots depict gene set enrichment, with colours indicating p-values and dot size corresponding to the number of overlapping genes between the list and the term/pathway (Count); the x-axis corresponds to the ratio of genes in the input list to genes in the term/pathway (GeneRatio). Scrmbl scramble, GO Gene Ontology, BP Biological Process, MF Molecular Function, HM Hallmark gene set, SynGO Synaptic Gene Ontologies, Con, scramble-injected control (bright-green), KD shRNA-injected knockdown (pale-green). Schematic in panel (**a**) was created in BioRender. Rocks, D. (2025) https://BioRender.com/oo9h37f.

female-biased chromatin changes by taking a broader sample of group-by-sex interaction regions ($p_{nominal} < 0.05$, fold-change > 1.5, Supplementary Data 8b) and filtering them further using divisive hierarchical clustering (Fig. 6a). This analysis revealed 5 total clusters, with the first two clusters containing 864 total regions (287 and 577 regions in Cluster 1 and 2, respectively) that lose accessibility specifically in females following Egr1 knockdown. Importantly, these regions

are annotated to genes enriched for synapse and dendrite-related terms (Fig. 6b), further implicating Egr1 in sex-dependent regulation of synaptic genes. Taking the 1165 total regions changed in females (301 sex-invariant, 864 female-specific), we found that nearly half of the genes annotated to these regions (46.4%, 540/1077) gain accessibility in the proestrus phase and after overexpression of Egr1 in OVX females (Fig. 6c). Enrichment analysis on these overlapping genes

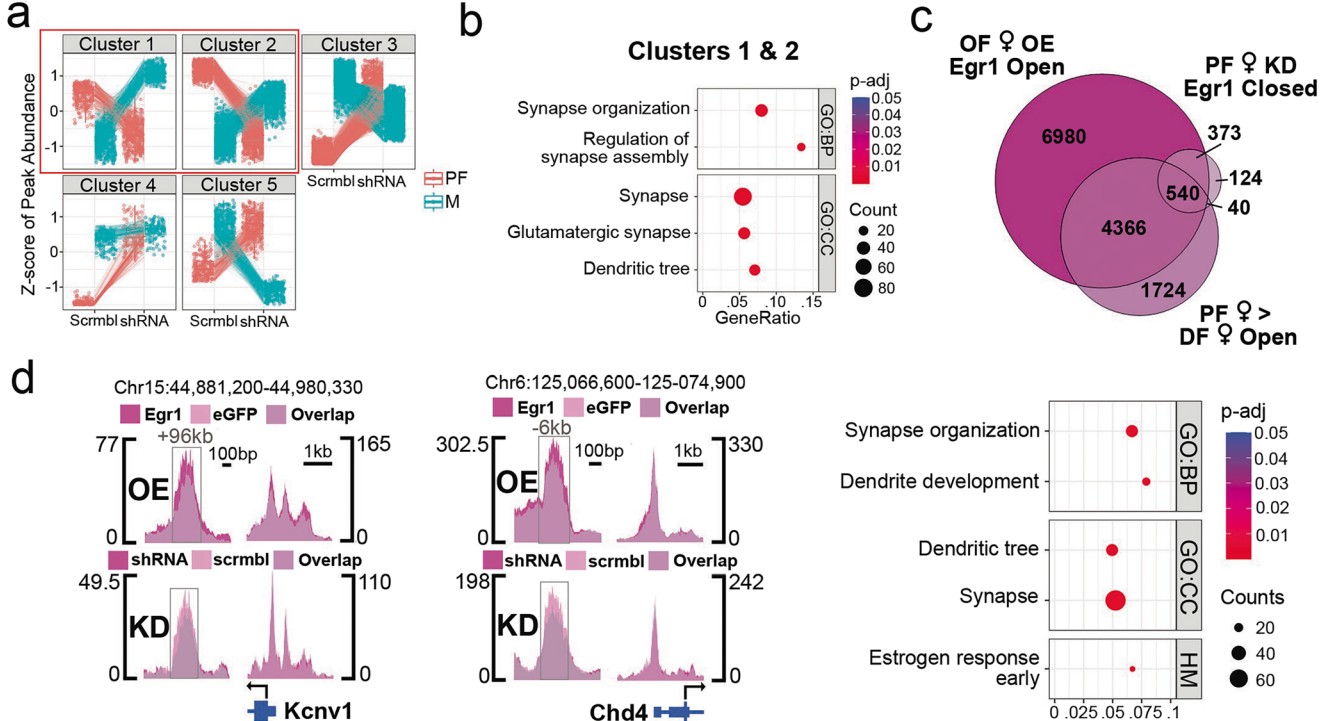

**Fig. 6 | Egr1 knockdown produces a dioestrus-like chromatin state in proestrus females. a** Divisive hierarchical clustering of differentially accessible regions (DARs) identified in the group-by-sex interaction analysis ($p_{nominal} < 0.05$, fold-change > 1.5) revealed 5 clusters of regions with differing effects of Egr1 knockdown in males (M) and proestrus females (PF). **b** Clusters 1 & 2 (boxed in **a**) correspond to DARs that lose accessibility following knockdown specifically in females, and genes annotated to these DARs are enriched for synapse related gene ontology (GO) terms. **c** 46.4% of genes associated with DARs that lose accessibility in proestrus females following Egr1 knockdown are similarly less accessible in ovariectomized females (OF) prior to Egr1 overexpression and in dioestrus females (DF) compared to proestrus females (PF; top). These overlapping genes are enriched for synapse and oestrogen response-related gene sets (bottom). **d** Two example genes include *Chd4* and *Kcnv1*, both of which are associated with chromatin regions that contain Egr1 motifs and become more accessible in OF following Egr1 overexpression and less accessible in PF following Egr1 knockdown. ATAC-seq data is shown using SparK plots of group-average normalized ATAC-seq reads ($n = 4$ biological replicates/group for overexpression data, $n = 3$ biological replicates/group for knockdown data). ATAC-seq data were analysed with the Wald statistical test (group effect; two-sided) or likelihood ratio test (group*sex effect) in DESeq2. Enrichment analysis was performed using hypergeometric tests. Exact *p*-values for the overexpression and knockdown ATAC-seq data can be found in Supplementary Data 4 and Supplementary Data 8, respectively. Dot plots depict Gene Ontology (GO) enrichment, with colours indicating p-values and dot size corresponding to the number of overlapping genes between the list and the term/pathway (Count); the x-axis corresponds to the ratio of genes in the input list to genes in the term/pathway (GeneRatio). OE overexpression, KD shRNA knockdown, PF proestrus females (red), M males (green), Egr1 Egr1-injected ovariectomized females (purple), eGFP eGFP-injected ovariectomized females (pink), shRNA shRNA-injected proestrus females (purple), scrmbl scramble-injected proestrus females (pink), BP Biological Process, CC Cellular Component, HM Hallmark gene sets.

revealed that Egr1 knockdown partially recapitulates a dioestrus-like chromatin state by reversing the Egr1-mediated gained-accessibility of neuronal chromatin around a subset of genes related to synapse regulation, dendrites, and oestrogen response. Two illustrative examples are *Kcnv1* (Supplementary Data 8b), encoding a voltage-gated potassium channel subunit, and *Chd4* (Supplementary Data 8a), encoding a chromatin remodelling enzyme (Fig. 6d). Both genes are associated with a chromatin region that contains an Egr1 binding motif and gains accessibility in OVX females following Egr1 overexpression while losing accessibility in proestrus females following Egr1 knockdown (Fig. 6d). Interestingly, *Kcnv1* is downregulated and *Chd4* is upregulated in proestrus compared to dioestrus females[18], which further supports the notion that Egr1 mediates downregulation of ion channels and upregulation of transcriptional regulators during the proestrus phase of the oestrous cycle (Fig. 4e). Together, our data indicate that Egr1 is required to maintain the proestrus chromatin state in vHIP neurons of cycling females, further supporting the role of this IEG in cyclical chromatin control relevant to female-specific neuronal and behavioural plasticity.

## Discussion

In summary, our study reveals that Egr1 is a sex-dependent IEG, capable of initiating and maintaining neuronal chromatin accessibility sex-

dependently, with consequences for targeted gene expression changes, structural plasticity, and anxiety- and depression-related behaviour. These findings are critical for our fundamental understanding of brain plasticity across sexes, but also have important implications for sex-biased brain disorders, anxiety, and depression disorders in particular.

IEGs have been known as major molecular adaptors that are rapidly induced by various extracellular stimuli, including growth factors, neurotransmitters, and hormones, allowing neurons to respond to an ever-changing environment[17]. Many IEGs are transcription factors and thus they provide a bridge that connects synaptic neuronal activity to long-term changes in gene expression underlying adaptive neuronal changes including synaptic and behavioural plasticity. While earlier studies emphasized epigenetic regulation of IEGs as part of their activity-dependent regulation[41–44], evidence for the role of IEGs as direct epigenetic regulators of their *target genes* is a newer finding[16,45,46] that provides mechanistic insight into IEGs' regulatory role in neuronal adaptation and plasticity. For instance, here we show the role of Egr1 in directing neuronal chromatin accessibility at its target genes, consistent with its previously established epigenetic regulatory role in recruiting Tet1 to shape the brain methylome in response to neuronal activity[40]. In addition, this is reminiscent of the role of cFos, another IEG, shown in males to direct chromatin opening

in a neuronal activity-dependent manner[16], although whether this occurs in the female brain is not known. Importantly, however, our study functionally links an IEG's effect on neuronal chromatin to structural and behavioural plasticity.

In addition, our data demonstrate that IEGs such as Egr1 allow for sex-dependent, sex hormone-driven chromatin and brain plasticity. While ovariectomy allowed us to isolate the effect of ovarian hormones in females, it also "stripped" sex as a variable of its major component and precluded using sex as a factor in our statistical analysis of the overexpression experiments. However, the knockdown experiment provided a complementary approach in cycling females, allowing for analysis across sex in intact male and female mice. With the multi-faceted approach that includes the integration of physiological, pharmacological, and gene manipulation data within the ovarian hormone-depleted (overexpression) and naturally-cycling (knock-down) background we provide robust evidence that the effects of Egr1 are indeed sex-dependent.

In fact, our results show that the cyclicity of ovarian hormone levels may provide a critical, sex-specific signal for Egr1's effect on chromatin regulation in females. Egr1's response to ovarian hormones seems to be optimized by oestrogen's regulation of Egr1 expression, as reflected in the cyclical Egr1 expression pattern across the oestrous cycle in the vHIP, but also in other brain regions such as the prefrontal cortex[47] or peripheral tissues such as uterus[48], as well as in the restoration of proestrus-like vHIP Egr1 levels observed when we treated OVX females with cyclical oestradiol regimen. Egr1's transcriptional role in brain regulation across the oestrous cycle was also reported by another group examining the prefrontal cortex in rats[47], indicating that Egr1 may be an oestrous-cycle-dependent chromatin regulator across brain regions in rodents. Importantly, we show that Egr1 expression is required for both the induction and maintenance of accessibility at Egr1 motifs. Studies by us and others also indicate that oestrogen regulates Egr1 expression via membrane-bound oestrogen receptors[18], which are abundant in the hippocampus, and can induce rapid effects via the activation of various kinase pathways[49]. However, the receptor signalling pathway leading to Egr1 induction remains to be established, and considering the hormonal complexity of the oestrous cycle, multiple hormones and receptors may be regulators of cyclical Egr1 levels.

It is also worth noting that, beyond Egr1's cyclical expression, we previously found a similar cyclic gene expression pattern for other epigenetic regulators, including multiple ATP-dependent chromatin remodelling factors, in vHIP neurons across the oestrous cycle[18]. Thus, cycling transcriptional regulators appear to be a distinctive feature of the oestrus cycle-driven gene regulation[18] that is likely to establish "the oestrous rhythm" in the cell. In fact, this cyclical expression in females may also be indicative of Egr1's role in driving the biological clocks in the body more generally, since Egr1 is also induced in the suprachiasmatic nucleus by light stimulation[50], and has been impli-cated in circadian rhythm-driven brain plasticity[21,51]. While the long-lasting viral manipulations of Egr1 levels performed here gave impor-tant insights into Egr1's sex-dependent effects on gene regulation and behaviour, the development of new molecular tools that allow for finer temporal control of gene expression will be important for future stu-dies to model the cyclical expression of Egr1 and other epigenetic regulators involved in driving the proestrus state.

In contrast to the rhythmic expression of ventral hippocampal Egr1 observed in females, we observe high basal expression of Egr1 in males at levels similar to proestrus females, consistent with our pre-vious study[18]. This may account for higher male baseline expression of genes such as Atp1a2, Ppp3r1, and Nrn1, and a lack of an Egr1 effect on the expression of these genes and related phenotypes, such as anxiety-related behaviour, due to a possible ceiling effect in males. However, for other phenotypes, such as Grin2a expression and dendritic spine density, which appear more similar between males and low oestrogenic (dioestrus or OVX) females, higher baseline Egr1 levels did not prevent phenotypic changes (Grin2a expression and spine increases) by Egr1 overexpression in males. These findings explain why we see extensive chromatin and gene expression changes in both sexes with partial overlaps in molecular and structural plasticity, indicating sex-dependent but not female-exclusive effects by Egr1.

Finally, in both males and females, IEGs are responders to various environmental factors of relevance to neuropsychiatric conditions and can be altered in various disorders[52]. Specifically, neuronal Egr1 expression has been shown to vary with early-life and social isolation stress[53,54], drugs of abuse including psychostimulants and opioid drugs[55], psychiatric disease including depression and schizophrenia[56,57], and with antidepressant treatment[58]. Considering its sensitivity to sex hormone levels and sex specificity of its actions, Egr1 therefore repre-sents a master molecular adaptor at the interface of the external and internal environment, with a critical role in sex-dependent neurobiology and psychiatric risk.

Importantly, we were able to go from genomics data generated using the physiological mouse model as a function of the oestrous cycle to in vivo mouse manipulation studies to validate a sex-dependent regulator of chromatin, neuroplasticity and behaviour. Since ovarian hormone shifts are an established but underexplored psychiatric risk factor in humans[3,11,12,59–61], our study represents an important contribution toward the discovery of sex-based drug targets and treatments for neuropsychiatric disorders.

## Methods

### Animals

For this study, we used seven separate cohorts of animals. The first cohort was used for behavioural and gene expression analysis of intact females across the oestrous cycle (Fig. 1b, c, Supplementary Fig. 1) and was comprised of $n = 48$ C57BL/6 J female mice that arrived at 5 weeks of age from Jackson Laboratory. The second cohort was used to compare the behaviour of gonadally intact females and ovar-iectomized (OVX) females that underwent ovariectomy at 4 weeks of age at the Jackson Laboratory (Supplementary Fig. 3) and included $n = 34$ intact and n = 12 OVX C57BL/6 J female mice. OVX mice were allowed to recover for one week before arriving, together with the intact females, at 5 weeks of age. After habituating for two weeks (5–7 weeks old), intact females from these cohorts underwent oestrous cycle tracking daily in the morning (between 9AM and 11AM) for two weeks (7-9 weeks) in order to establish a predictive cycling pattern for each female animal and to ensure that only females with regular cycles were included in the study (see Oestrous cycle monitoring). Vaginal smears were also taken from OVX females for 4 days to verify that ovariectomy was successful. This was followed by behavioural testing at 9–11 weeks of age (see Behavioural testing). Animals were sacrificed at 11 weeks of age, brains were extracted, and bilateral ventral hippo-campi were dissected on ice then flash frozen in liquid nitrogen.

The third cohort of mice was used for cyclical oestrogen treat-ment experiments (Fig. 1d–f) and was comprised of $n = 32$ C57BL/6 J female mice that arrived at 6 weeks of age from Jackson Laboratory. Following two weeks of habituation, a subset of mice from this cohort underwent either ovariectomy ($n = 16$) or sham surgery ($n = 8$) at 8 weeks of age in-house at Fordham University (see Ovariectomy sur-gery). After recovering from surgery for 1 week, OVX mice were cycli-cally treated with either corn oil vehicle ($n = 8$) or 1 μg oestradiol benzoate (EB) in corn oil ($n = 8$) via subcutaneous injection once every four days for a 6-week treatment period. As above, vaginal smears were taken from OVX females to confirm ovariectomy, while the oestrous cycle of intact control ($n = 8$) and sham females was tracked briefly to ensure proper cyclicity. Mice from this cohort were tested in the open field (see Behavioural testing) at 14–15 weeks of age and sacrificed at 15 weeks of age. Brains were extracted, and bilateral ventral hippo-campi were dissected and flash frozen in liquid nitrogen. For the

analysis in this cohort, proestrus mice were merged from sham and control females as no effect of sham surgery of relevance to this study was identified (Supplementary Fig. 12). Intact mice that were not in the proestrus stage, as confirmed by vaginal smear cytology, at the time of behavioural testing or sacrifice were excluded from the relevant analyses.

The following three cohorts (Cohorts 4-6) were used in the Egr1 overexpression experiments (Figs. 1g–i, 2–4, and Supplementary Fig. 2, 4–11) and included OVX C57BL/6 J females alongside age-matched males. These cohorts were used for behaviour ($n$ = 24/sex), molecular analyses ($n$ = 20/sex), and Golgi staining ($n$ = 10/sex), respectively. Females underwent ovariectomy at the Jackson Laboratory at 4 weeks of age, just prior to the onset of puberty, to ensure that females used in these experiments never experienced ovarian hormone cycling and do not retain a "cellular memory" of oestrous cycle-related processes. Females recovered for 1 week after the surgery at the Jackson Laboratory, and arrived at Fordham University at 5 weeks of age, together with age-matched males. The final, seventh cohort was used in the Egr1 knockdown experiments (Figs. 5, 6, and Supplementary Fig. 10) and included intact C57BL/6 J females and age-matched males ($n$ = 6/sex) that arrived at Fordham University at 5 weeks of age. All animals in Cohorts 4–7 were allowed to habituate for two weeks before undergoing stereotaxic surgery (see *Stereotaxic surgery*) at 7 weeks of age and were allowed to recover for ~3 (overexpression) or 4 (knockdown) weeks to ensure stable transgene expression. During recovery, 4 days of vaginal smears were obtained from OVX females to confirm ovariectomy. The oestrous cycles of intact females used in the knockdown experiments were tracked during the final two weeks of recovery to ensure that all females were sacrificed during the proestrus phase. In the behaviour Egr1 overexpression cohort, animals underwent behavioural testing from 10 to 11 weeks of age before being sacrificed at 11 weeks of age, at which time the whole brain was removed and preserved for cryosectioning (see *Immunohistochemistry*) so that we could verify viral targeting in every animal that underwent behavioural testing. For the molecular analysis Egr1 overexpression and knockdown cohorts, animals were sacrificed at 10–11 weeks and bilateral hippocampi were dissected on ice and flash frozen in liquid nitrogen. Analysis of RNA-seq data in these cohorts was used to confirm overexpression or knockdown of Egr1. For the Golgi staining Egr1 overexpression cohort, animals were sacrificed at 10 weeks, and the whole brain was removed then underwent Golgi-Cox staining (see *Golgi-cox staining for analysis of dendritic spines*).

All animals used in this study were housed in same-sex cages (n = 3-5/cage) and were kept on a 12:12 h light:dark cycle with *ad libitum* access to food and water. The temperature of the room is maintained at 21 °C, and the humidity ranges from 30 to 70%. All mouse tissue was generated from mice euthanized by cervical dislocation followed by rapid decapitation and dissection of either whole brains or ventral hippocampal tissue. All frozen tissues were stored at −80 °C before further processing. All animal procedures were approved by the Institutional Animal Care and Use Committee at Fordham University.

## Oestrous cycle monitoring

Oestrous cycle tracking was performed using vaginal smear cytology, a well-established method, as described previously in refs. 18,62,63. Smears were collected by filling a disposable transfer pipette with 100 μl of distilled water, gently placing the tip of the pipette at the vaginal opening, and collecting cells via lavage. The cell-containing water was then applied to a microscope slide and allowed to dry at room temperature for two hours. Once dried, slides were stained with 0.1% crystal violet in distilled water, washed, and then allowed to dry prior to examination with light microscopy. Oestrous cycle stage can be determined by the relative quantities of nucleated epithelial cells, cornified epithelial cells, and leukocytes[18]. The proestrus phase is characterized by mostly nucleated epithelial cells, while the oestrus phase is characterized by mostly cornified epithelial cells. Metestrus and dioestrus have both forms of epithelial cells as well as leukocytes, and dioestrus exhibits a higher proportion of leukocytes compared to metestrus. After determining the oestrous cycle stage of each female animal daily for two weeks, or three cycles, cycling patterns can be established and stage predictions can be made for the grouping of animals for behavioural tests and molecular analyses. Grouping predictions were confirmed by obtaining vaginal smears of animals following behavioural testing or sacrificing. We previously validated that our predictions correspond to the expected ovarian hormone content in both serum and hippocampal tissue for proestrus (high-oestrogen, low-progesterone) and dioestrus (low-oestrogen, high-progesterone) mice[18]. Animals with irregular cycles were excluded from behavioural and molecular analyses.

## Ovariectomy surgery

In the third cohort of mice (see *Animals*), ovariectomy was performed at 8 weeks of age in house[20]. Anaesthesia was achieved with isoflurane using an induction chamber and maintained with a continuous flow of isoflurane (1.5–2%). Aseptic technique was utilized throughout the procedure, which involved a bilateral skin incision followed by excising the exposed ovary at the tip of the uterine horn on each side. Sham surgeries involved identical incisions, as well as manipulation and exposure of the ovaries, but lacked the excision, leaving the ovaries intact. Mice were closely monitored for 3–4 days following surgery to ensure adequate recovery, were administered carprofen for 2 days, and were given Nutra-Gel (Bioserv, S4798) food packs to limit the activity required to obtain food during this recovery period.

## Candidate gene expression

To analyse the gene expression of candidate genes, RNA was isolated from the ventral hippocampus with Qiagen's Allprep DNA/RNA Mini Kit. Reverse transcription was then performed with Invitrogen's SuperScript III First-Strand Synthesis kit, and qPCR was performed with Applied Biosystems' FAST SYBR Green and the Quant Studio 3 PCR machine. Relative *Egr1* and *Ppp1r1b* transcript levels were determined using the $2^{-\Delta\Delta Ct}$ method with Cyclophilin A (*Ppia*) as the endogenous reference gene. The following primer sequences were used in this analysis: *Egr1* (forward: 5′- AGCGAACAACCCTATGAGCAC-3′, reverse: 5′-GGATAACTCGTCTCCACCATCG-3′); *Ppp1r1b* (forward: 5′-GGACGAAGAAGAAGACAGCCA-3′, reverse: 5′-CACTTGGTCCTCAGAGTTTCC-3′); *Ppia* (forward: 5′-GAGCTGTTTGCAGACAAAGTTC-3′, reverse: 5′-CCCTGGCACATGAATCCTGG-3′).

## Behavioural testing

Animals underwent behavioural testing from 9 to 11 weeks of age (Cohorts 1, 2 & 4) or at 14–15 weeks of age (Cohort 3; see *Animals*). For all behavioural tests, the movement of animals was tracked with a camera and AnyMaze© tracking software. For the open field test, animals were placed in the corner of a 40 × 40 x 35 cm open field arena (Stoelting Co.). The animal's movement was tracked for 10 min, and the total distance travelled by the animal, the time spent in the centre versus the periphery of the maze, and the number of entries into the centre were recorded. For the elevated plus maze test, animals began the test by being placed in the centre of the raised (40 cm) plus-shaped platform (two 35 x 5 cm arms; Stoelting Co.). The movement of the animal was tracked for 5 min, and the total distance travelled by the animal, as well as the time spent in the open arms versus the closed arms was recorded. For the forced swim test, animals were placed in a 2 L glass beaker with 1 L of lukewarm water. Movement was tracked for 2 minutes and the time spent immobile was recorded.

## Adeno-associated viruses

Recombinant adeno-associated viruses (AAVs) were used to overexpress or knock down Egr1 in ventral hippocampal neurons. The

AAV9 serotype was selected due to its suitability for gene transduction in brain tissue[64]. The two overexpression AAVs used in these experiments were an experimental AAV and a control AAV; the experimental AAV contains the Egr1 transgene, an IRES element, and eGFP, with the IRES element ensuring coproduction of Egr1 and eGFP from a single AAV, while the control AAV contains only eGFP. The knockdown AAVs contain either an shRNA targeting Egr1 (experimental AAV) or a scrambled shRNA (control AAV), with each coexpressing eGFP. All AAVs utilize the neuron-specific hSYN promoter[65] and contain a WPRE enhancer used to increase the expression of the respective transgenes, with the experimental overexpression AAV containing a smaller iteration of this enhancer (WPRE3) due to space limitations, which retains a high degree of enhancer activity[66]. All viruses were purchased from Vector Biolabs (overexpression: AAV-258146, eGFP control: 7076; knockdown: shAAV-258146, scramble control: 7045) and were provided in PBS solution with 5% glycerol at a titre of ~$10^{13}$ genome copies/ml.

## Stereotaxic surgery

Animals were anaesthetized prior to surgery using isoflurane delivered to an anaesthesia induction chamber and kept anaesthetized during surgery with a continuous flow of 1.5–2% isoflurane. Either control or experimental AAV solution (overexpression: $n = 12$/group/sex for behaviour, $n = 6$/group/sex for RNA-seq, $n = 4$/group/sex for ATAC-seq, $n = 5$/group/sex for Golgi staining; knockdown: $n = 3$/group/sex for RNA-seq and ATAC-seq) was delivered to the injection site through a glass pipette (15 μm diameter) fixed to a Nanoject III© microinjector pump. The coordinates (relative to bregma) used for ventral hippocampal injections were: A/P: −2.95, M/L: +/− 2.85 (bilateral injections), D/V: −3.8, −3.9. 200 nL was delivered (10 pulses of 20 nL with 15 seconds in between) at D/V −3.9, then again at D/V −3.8 totalling 400 nL of injection/side. Three (overexpression) or four (knockdown) weeks were allowed to pass to permit stable expression levels of the viral transgenes before behavioural testing was conducted or before the animals were sacrificed for molecular analyses or Golgi staining. In order to verify proper viral delivery into the ventral hippocampus of the animals that underwent behavioural testing, immunostaining of brain sections was performed (see *Immunohistochemistry*). Animals were excluded if there was no viral expression present or if there was evidence of off-target expression or tissue damage (this occurred in $n = 4$ females and $n = 2$ males, or 12.5% of animals tested). For molecular analyses, we were able to confirm efficient Egr1 overexpression or knockdown in ventral hippocampal neurons using Egr1 transcript levels (RNA-seq) and chromatin accessibility data (ATAC-seq).

## Immunohistochemistry

Fresh dissected brains were washed with ice-cold 0.1 M PBS and fixed in 4% PFA in 0.1 M PBS at 4 °C for 24 h. After fixation, brains were rinsed in cold 0.1 M PBS and underwent sucrose preservation, which involved placing the brains in solutions containing 15% then 30% sucrose dissolved in 0.1 M PBS at 4 °C for 24 h and 48 h, respectively. Brains were then frozen in dry ice-cooled hexane and stored at -80 °C until sectioning. Cryosectioning was performed by embedding brains in optimal cutting temperature compound (OCT) and cutting serial sections on a rotary cryostat (Leica CM1950, Leica Biosystems GmBH). 20 μm coronal sections containing the ventral hippocampus were collected on Super Frost Ultra Plus slides (Fisher Scientific) and stored at −80 °C prior to immunostaining. Immunostaining involved rehydrating slides in 0.1 M PBS for 30 min at room temperature. After rehydration, blocking buffer (5% BSA and 0.4% Triton X-100 in 0.1 M PBS) was applied for 1 hour at room temperature. Slides were then washed with PBS-T (1% BSA and 0.4% Triton X-100 in 0.1 M PBS), and the Egr1 primary antibody was applied (Rabbit anti-Egr1 mAb Cell Signal #4153, 1:500 in PBS-T) for 24 h at 4 °C. Following primary antibody incubation, slides were washed with PBS-T, then the secondary antibody was applied (Donkey anti-Rabbit IgG pAb Invitrogen A-21207, 1:250 in PBS-T) for 2 hours in the dark at room temperature. Following secondary antibody incubation, slides were washed with PBS-T, then DAPI was applied (1:1000 in PBS) for 5 minutes in the dark at room temperature. Following DAPI incubation, slides were washed with 0.1 M PBS and mounted with Mowiol 4-88 and a coverslip. For all staining sessions, sections with no primary antibody (PBS-T applied instead of primary antibody) and sections with no secondary antibody (PBS-T applied instead of secondary antibody) were included to ensure that the observed fluorescent signal corresponded to Egr1 detection rather than autofluorescence of the tissue or non-specific binding of the secondary antibody.

## Fluorescence-activated nuclei sorting (FANS)

For overexpression and knockdown experiments, all molecular analyses were performed on neuronal nuclei, which were purified using FANS, as described previously in refs. 67,68. Briefly, for the overexpression ATAC-seq experiment, 4 animals per group (eGFP, Egr1) per sex were included (total $n = 16$). For the overexpression nucRNA-seq experiment, 6 animals per group (eGFP, Egr1) per sex were included (total $n = 24$). The overexpression ATAC-seq experiment used bilateral ventral hippocampi from a single animal for each biological replicate ($n = 4$/group/sex), while for overexpression nucRNA-seq, bilateral ventral hippocampi were pooled from two animals for each biological replicate ($n = 3$ replicates/group/sex). For the knockdown ATAC-seq and nucRNA-seq experiments, 3 animals per group (scrmbl, shRNA) per sex were included (total $n = 12$) with each biological replicate comprised of bilateral ventral hippocampi from a single animal. We performed FANS in batches of 3–4 biological replicates per session and ensured that the groups were evenly dispersed across batches for each experiment. Brain tissue was dissociated in lysis buffer using a tissue douncer, and nuclei were isolated from tissue lysates by ultracentrifugation through a sucrose gradient[67]. Pelleted nuclei were resuspended in DPBS and incubated for 45 min with the mouse monoclonal antibody against neuronal nuclear marker NeuN conjugated to AlexaFluor 488 (1:1000, MAB377X; Millipore, MA; clone A60). Before sorting, we added DAPI (1:1000) to the incubation mixture and filtered all samples through a 35 μm cell strainer. FANS was performed on a FACSAria instrument (BD Sciences), and data were collected and analysed using BD FACSDiva v8.0.1 software at the Albert Einstein College of Medicine Flow Cytometry Core Facility (Supplementary Fig. 13). In addition to a sample containing NeuN-AlexaFluor 488 and DAPI stain, three controls were used to set up the gates for sorting: DAPI only; IgG1 isotype control-AlexaFluor 488 (1:1000, Mouse monoclonal IgG1-k, FCMAB310A4; Millipore, MA; Clone MOPC-21) and DAPI; and NeuN-AlexaFluor 488 only. We set up the protocol to remove debris, ensure single nuclear sorting (using DAPI), and select the NeuN+ (neuronal) and NeuN- (non-neuronal) nuclei populations (Supplementary Fig. 13). For the ATAC-seq experiments, we collected 50,000 NeuN+ nuclei per biological replicate in BSA-precoated tubes filled with 200 μL of DPBS. For the nucRNA-seq experiments, we collected 50,000–250,000 nuclei directly into Trizol LS reagent (Thermo Fisher Scientific, 10296-010) to protect RNA from degradation. The nuclei used for knockdown nucRNA-seq were the remaining nuclei from the same tissue used for ATAC-seq.

## nucRNA-seq

Following sorting of nuclei, chloroform was added to the sample, and the aqueous phase was recovered. nucRNA was then isolated and purified using the RNeasy Micro kit (Qiagen). The quality of nucRNA was assessed using the Fragment Analyzer (Agilent) or Bioanalyzer (Agilent), with each sample having RQN > 7.1 (7.9 average) or RIN > 6.2 (7.5 average) (Supplementary Fig. 14a, Supplementary Data 10). nucRNA quantity was determined using the Qubit RNA High-Sensitivity assay (Thermo Fisher Scientific, Q32852). All purified nucRNA

(3–26 ng/sample) was used to construct cDNA sequencing libraries using the KAPA RNA HyperPrep Kit with RiboErase (KK8560, KAPA Biosystems), following the manufacturer's instructions. rRNA was depleted from nucRNA samples using hybridization of DNA oligonucleotides complementary to rRNA followed by RNase H and DNase treatment. The efficiency of rRNA depletion was confirmed using the Bioanalyzer (Agilent) and with qRT-PCR amplification of the 28S rRNA transcript before and after RiboErase treatment using the following primers: Forward, 5′- CCCATATCCGCAGCAGGTC-3′; Reverse, 5′-CCAGCCCTTAGAGCCAATCC-3′. rRNA-depleted nucRNA was then fragmented at 94 °C for 5 minutes in the presence of Mg2 + , followed by first and second strand cDNA synthesis and A-tailing. After ligation of KAPA dual-indexed adapters (KK8722), each library was amplified using 13–15 PCR cycles. Following a bead-based clean-up, the length of the libraries (368 bp, on average, Supplementary Data 10) and their quality was assessed using the Bioanalyzer High-Sensitivity DNA Assay (Supplementary Fig. 14b). The libraries were quantified with the Qubit HS DNA kit (Life Technologies, Q32851) and by qPCR (KAPA Biosystems, KK4873) prior to sequencing. 150 bp, paired-end sequencing was performed in one lane of an S1 flow cell on the NovaSeq 6000 instrument at the New York Genome Center, yielding 52–99 million reads per sample (Supplementary Data 10).

## RNA-seq analysis
Sequences were adapter-trimmed and aligned to the mouse reference genome (mm39) with the GENCODE M32 gtf (overexpression) or M36 gtf (knockdown) file using Star Aligner[69]. The gene counts produced by Star were collated into a matrix. Differential gene expression analysis was then performed using DESeq2[70] at significance level $p_{adj} < 0.05$. For the overexpression experiment, ovariectomy in females precluded a direct comparison across sex, and we therefore analysed the effect of group (Egr1 vs. eGFP) separately in males and ovariectomized females using the Wald statistical test. For the knockdown experiment, use of intact females and males allowed us to include sex as a factor in our analysis; we therefore analysed the effect of group (shRNA vs. Scramble) in males and proestrus females together, with sex included as a factor in the model (- Group + Sex; Wald test), and also performed a group-by-sex interaction analysis (- Group + Sex + Group:Sex; likelihood ratio test). Counts for the knockdown experiment were normalized using the endogenous reference gene *Ppia*. Volcano plots were created using EnhancedVolcano (https://github.com/kevinblighe/EnhancedVolcano). Enrichment analyses were performed using FUMA[71] at significance level $p_{adj} < 0.05$. Enrichment analysis was also performed using GSEA[72] with ranked gene lists and results were plotted using the EnrichmentMap[73] and AutoAnnotate apps in Cytoscape[74].

## ATAC-seq
We performed ATAC-seq according to Buenrostro et al.[75] and our previously published protocol on sorted neuronal nuclei[18,68], with some modifications[76]. Following FANS, nuclei were pelleted and resuspended in 50 µL of the transposase reaction mix including 25 µL of the 2xTD reaction buffer, 5 µL of transposase enzyme (Illumina Tagment DNA TDE1 kit, 20034197). We also included additional components: 16.5 µl 0.1 M PBS, 0.5 µl 1% digitonin, and 0.5 µl 10% Tween-20, which were previously shown to enhance transposition efficiency. Transposition occurred at 37 °C for 30 minutes and transposed DNA was purified using the MinElute PCR Purification Kit (Qiagen, 28004). Indexing libraries using Nextera i5 and i7 indexed amplification primers (Illumina XT index kit, 15055290) occurred alongside library amplification with NEBNext High-Fidelity PCR Master Mix (New England Biolabs, M0541S). The PCR reaction was carried out with the following conditions: 1 cycle of 72 °C for 5 min and 98 °C for 30 s; followed by 10 cycles of 98 °C for 10 s, 63 °C for 30 s, and 72 °C for 1 min. Amplified libraries were then purified using the MinElute PCR

Purification Kit. Library quality was then assessed using the Bioanalyzer High-Sensitivity DNA Assay (Supplementary Fig. 14c). The ATAC-seq libraries were quantified by Qubit HS DNA kit and by qPCR prior to sequencing. 150 bp, paired-end sequencing was performed in an S1 flow cell on the NovaSeq 6000 instrument at the New York Genome Center, yielding 40–150 million paired-end reads per sample (Supplementary Data 11).

## ATAC-seq analysis
Sequences were adapter-trimmed and aligned to the mouse reference genome (mm39) using BWA-MEM software[77]. Duplicates and mitochondrial reads were removed, then fragment length distribution plots were generated to confirm that insert sizes corresponded primarily to nucleosome-free, mono- and di-nucleosomal DNA (Supplementary Fig. 14d). Next, peak-calling was performed using MACS2[78] as previously reported[75] and called peaks were filtered using the mm39 blacklist[79]. We ensured all samples had ≥ 20% of reads in peaks, indicative of high-quality ATAC-seq samples (Supplementary Data 11). Differential accessibility of open chromatin regions was assessed using DiffBind as described previously[80]. Briefly, reads were counted in consensus peaks (shared by at least two replicates), normalized, then quantitatively compared using DESeq2[70] with a significance threshold of $p_{adj} < 0.05$ for the effect of group. The DESeq2 analysis of ATAC-seq data followed the same strategy as the RNA-seq analysis, with overexpression data analysed by evaluating the effect of group separately in males and ovariectomized females. Then, for the knockdown data, we analysed the effect of group together in males and proestrus females (-Group + Sex), along with a group-by-sex interaction (-Group + Sex + Group:Sex) analysis. To identify sex-specific patterns of chromatin accessibility in the knockdown experiment, we applied unbiased clustering to a broad group of DARs as described previously[81]. Specifically, we clustered group-by-sex interaction DARs ($p_{nominal} < 0.05$ and fold-change > 1.5) using DEGreport (https://lpantano.github.io/DEGreport/) which utilizes the DIANA divisive clustering algorithm[82]. Annotation of differentially accessible regions (DARs) was performed using ChIPSeeker[83] on the TxDb.Mmusculus.UCSC.mm39.refGene annotation, and we identified overlapping peaks using the findOverlapsOfPeaks function of ChIPpeakAnno[84]. Motif analysis was performed using HOMER[85] (hypergeometric test), and enrichment analysis on lists of annotated genes was performed with FUMA[71] (hypergeometric test) and EnrichR[86] (hypergeometric test) at a significance threshold of $p_{adj} < 0.05$. Plots of aligned ATAC-seq data were created using SparK[87]. Heatmaps and profile plots of H3K4me1, H3K27ac, and H3K4me3 enrichment within gained-open (overexpression) or -closed (knockdown) Egr1 binding sites were performed using deepTools[88] using the publicly available ChIP-seq data GSE74964[27] generated using neuronal (NeuN + ) CA1 hippocampal neurons from male mice. The same deepTools analysis was performed for Egr1 ChIP-seq signal enrichment using the publicly available ChIP-seq data GSE108768[40] generated using prefrontal cortical tissue from male mice.

## Integration of genomics data
Our ATAC-seq and nucRNA-seq data previously generated from ventral hippocampal neurons (GSE114036) of proestrus female, dioestrus female, and male mice[18] were integrated with the data used in this study. For the purposes of overlapping the RNA-seq data, we applied the significance threshold used in the previous study[18] ($p_{nominal} < 0.05$). Plots of aligned ATAC-seq data were created using SparK[87].

## Golgi-cox staining for analysis of dendritic spines
Whole brains of mice injected with either control or Egr1 AAV (see *Adeno-associated viruses*, n = 5/sex/group) were rinsed with distilled water and underwent the Golgi-Cox staining procedure (Golgi-Cox OptimStain Kit, Hitobiotec Inc. #HTKNS1125) following the

manufacturer's instructions. Briefly, whole brains were submerged in premixed Golgi-Cox impregnation solution and stored in the dark at room temperature for 24 h, after which the solution was replaced and the brains remained submerged for an additional two weeks. Samples were then transferred to a tissue-protectant solution and kept in the dark at 4 °C for 12 h, after which the solution was changed and the brains remained submerged for an additional 72 h. Samples were then frozen in dry ice-cooled hexane and stored at −80 °C until sectioning. 100 μm sections of OCT-embedded brains were cut using a cryostat (Leica CM1950) and collected on gelatin-coated slides. After drying overnight, slides were stained according to the kit protocol and mounted using DPX (Sigma-Aldrich). Brightfield Z-stacks of dendrites from ventral hippocampal CA1 pyramidal neurons were taken at 60x magnification and analysed using ImageJ (https://imagej.nih.gov/ij/index.html).

## Statistics

Statistics were performed using SPSS and R. For the analysis of candidate gene expression and behaviour i) across the oestrous cycle (Fig. 1b, c; Supplementary Fig. 1), ii) in OVX females undergoing cyclical oestradiol or vehicle treatment alongside untreated proestrus females (Fig. 1e, f), and iii) in prepubertally ovariectomized females alongside intact proestrus and dioestrus females (Supplementary Fig. 3), one-way ANOVA was used with Holm's post hoc test. For behavioural and dendritic spine density analysis of animals from the overexpression experiments, Welch two-sample t-tests were performed separately in males and ovariectomized females, since ovariectomy precluded the analysis across sexes. The effect size for the dendritic spines experiment was reported for each sex as Cohen's d, calculated as the difference between the group means of Egr1 and eGFP groups divided by the pooled standard deviation. For each of these statistical tests, $p < 0.05$ was considered significant. Detailed results of all ANOVA and t-tests are available in Supplementary Data 1.

## Reporting summary

Further information on research design is available in the Nature Portfolio Reporting Summary linked to this article.

## Data availability

RNA-seq and ATAC-seq data from Egr1 overexpression and knockdown experiments are available from the NCBI Gene Expression Omnibus database under accession GSE249978. ChIP-seq data for H3K4me1, H3K27ac, and H3K4me3 in neuronal CA1 neurons of male mice were obtained from GSE74964[27]. ChIP-seq data for Egr1 in the prefrontal cortex of male mice was obtained from GSE108768[40]. RNA-seq and ATAC-seq data from vHIP neurons of proestrus females, dioestrus females, and males which we previously published[18] are available at GSE114036. Any other relevant data supporting the key findings of this study are available within the article and its Supplementary Information files or from the corresponding author upon request. A reporting summary for this article is available as a Supplementary Information file. Correspondence and requests for materials should be addressed to M.K. (mkundakovic@fordham.edu). Source data are provided with this paper.

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

## Acknowledgements

This work was supported by the National Institutes of Health award R01MH123523 (to M.K.). We would further like to acknowledge the resources of the Center for Epigenomics at the Albert Einstein College of Medicine. Finally, we would like to thank Ming Liu and Lara Winterkorn for their assistance with nuclei sorting and next-generation sequencing, respectively. Schematics in Figs. 1, 2, 3, and 5, and in Supplementary Figs. 1 and 6 were created using BioRender.com (Created in BioRender. Rocks, D. (2025) https://BioRender.com/oo9h37f).

## Author contributions

D.R. and M.K. designed the study; D.R., L.D., L.O., E.P., E.F.G., and M.K. performed experiments. D.R. and M.S. performed data analyses. D.R., H.C., and M.K. interpreted the data. D.R. constructed the figures. J.M.G. contributed computational resources. E.F.G. contributed to AAV vector design. D.R. and M.K. wrote the article. M.K. conceived and directed the project. All authors commented on and approved the final version of the paper.

## Competing interests

The authors declare no competing interests.
