## [Transparent Peer Review file · Nature Communications]

Egr1 is a sex-dependent regulator of neuronal chromatin, structural plasticity, and behaviour

Corresponding Author: Dr Marija Kundakovic

Version 0:

Reviewer comments:

Reviewer #1

(Remarks to the Author)

Overall, the authors have done an outstanding job of responding to my previous critiques, as well as those of my fellow Reviewers. The new data & analyses provided nicely support the authors' assertions, and the text revisions have greatly improved the clarity of the manuscript, along with providing more precise language (particularly with respect to the structural morphology experiments) so as to not over interpret their findings. I continue to feel that this manuscript is robust and will be of broad interest to the field. As such, I now support publication of this work at Nature Communications.

Reviewer #2

(Remarks to the Author)

The manuscript by Rocks et al describes the effects of overexpressing the transcription factor Erg1 in ventral hippocampus of male and female mice in studies testing the possibility that this IEG regulate chromatin structure and gene expression changes underlying synaptic plasticity and behavioral changes across the female estrous cycle. The original submission received three detailed and extensive reviews that brought up issues of experimental design, the strength of support for some of the conclusions reached and wording. The authors have gone to great lengths to modify the manuscript in response to the review comments including the addition of additional experiments and analyses of publicly available data (response to Reviewer #1), revision of statistical analysis and wording (response to all three reviewers), and changes in interpretation (all three reviewers). The revisions increased support for the conclusions reached and improved language so as to not overinterpret the results. Moreover, specific elements of experimental design were explained and some of the experimental findings were more clearly acknowledged. Overall, it is a strong manuscript, providing novel evidence for consequences of Erg1 overexpression in females which indicate mechanisms through which normal fluctuations in Erg1 expression across the estrous cycle in females may control gene expression.

Regarding response to Reviewer #1 point 3, I agree with the authors in that analysis of overexpression in non-OVX females could give rise to complex and difficult-to-interpret results. In those animals there would be a hugely different hormonal environment that could provide counter balancing influences.

The authors made a number of wording changes, reflecting review comments (e.g., the changes from 'synaptic plasticity' to 'structural plasticity' and from sex-specific to sex-dependent is important). Also important is the recognition, and now fuller description, that many of the effects of Erg1 overexpression are similar between males and females – this is now note and more fully considered in the manuscript.

I do think the authors are still selling their results a bit hard at the end of the abstract and discussion but the changes in wording are an improvement.

Line 95: I think "across sex" should be changed to "between the sexes" (also line 116)

Reviewer #3

(Remarks to the Author)

In their revised manuscript, the authors have added substantial new analyses and an entirely new Egr1 knockdown experiment as suggested by Reviewer 1. I appreciate that they provided the complete lists of their DEGs, GO terms and DARs for their genomic experiments formatted as Supplementary Tables. However, I still have concerns with the overall study design, interpretation and conclusions. As I stated previously, the Egr1 overexpression experiment does not effectively model Egr1 levels in proestrus. Partial rescue of spine and behavior phenotypes with Egr1 overexpression suggest that additional proestrus-dependent mechanisms beyond Egr1 expression contribute to increased spines and decreased anxiety-related behaviors during proestrus. Egr1 overexpression only captures a fraction of the authors' published proestrus-specific changes in gene expression and chromatin accessibility. Extended Data 9 shows very minimal overlap between DARs that open with Egr1 overexpression and DARs that change across the estrous cycle. Taken together, these results suggest that the Egr1 overexpression experiments are not modeling the physiological increase in Egr1 expression with proestrus. Overexpression will persistently boost Egr1 expression in all neuronal cell types as is apparent in Extended Data Figure 2. The mild effects of Egr1 knockdown experiments in proestrus additionally demonstrate that the overexpression experiments are a bit of a sledgehammer approach.

I dispute the claim that ovariectomy precludes the inclusion of sex as a variable. Female gonadectomy does not preclude the ability to perform statistical comparisons with males. The authors repeatedly use the term "sex-dependent" and their primary conclusion is that Egr1 acts differently in males and females. Accordingly, the Egr1 overexpression and control comparisons should be made in the males and the ovariectomized females in the same statistical analysis, using a 2 way ANOVA. This should be done throughout the manuscript whenever there are adjacent pairwise comparisons (spines, behavior, genes/DARs). Figure 4 shows that Egr1 overexpression in OVX females boosts spine numbers to similar numbers as in control and OE males. Reviewer 2 makes the same argument in their point 7.

Additional points:

In the Egr1 knockdown experiment, the authors claim that genes and DARs that decrease or close with Egr1 knockdown are enriched for estrogen response genes, but the Figures show minimal enrichment. Figure 5c shows perhaps 5 genes associated with the term "estrogen response early" and Figure 5g indicates between 7.5 and 10 genes located near regions that lose accessibility with Egr1. What are these genes and sites? Are the closing sites near the down-regulated genes? The downregulated genes Sh3bp5 and Rara do not have DARs annotated to them in Supp Table 8a.

Kcnv1 is shown in Figure 6, but there is no DAR associated with this gene in Supp Table 8a.

Extended Data 10 should include controls. The authors should plot signal from the published Egr1 ChIP-seq in all groups. For panel a), this includes the ATAC peaks for GFP control-injected males and females. For panel b), this includes ATAC peaks from the scrambled RNA controls and all phases of the estrous cycle.

In the publication for the Egr1 ChIP-seq data used here, it was reported that ~44% of binding sites are at promoters, while here the authors report that "neuronal enhancers are the primary substrate for targeted opening of chromatin by Egr1" in their overexpression experiments. Are the promoter sites not open in the ventral hippocampus, or are they simply not differential between groups?

The example ATAC-seq library in Supplementary Figure 2 looks under-transposed. Although a nucleosomal ladder is apparent, very little of the library consists of accessible chromatin, defined by Buenrostro et al (2013) as mono- and di-nucleosomes and subnucleosomal fragments. The authors should generate plots of their fragment distribution to ensure that the reads they aligned are predominantly fragments of accessible chromatin, as in Nature Methods papers PMID: 24097267 and PMID: 28846090.

Overstatements throughout the manuscript are distracting. It is not that Egr1 is "sex-dependent" but that its expression and consequent effects are increased in proestrus. It is likely that many IEGs show sex differences in expression due to sex differences in sex hormones and neural activity. Fos was first reported to show sex differences in the rat brain 20 years ago. PMID: 15842237

Additional overstatements include the following:

Line 52. "So far, molecular studies in neuroscience have focused on the male brain. Please omit this sentence. Many if not most modern molecular neuroscience studies include females, in line with SABV requirements from major funding institutions, and there has been extensive work over the past decade on mechanisms of estrogen signaling in the female rodent brain. The cited reference 14 is from 2011.

Lines 182-184: "Pathway analysis on genes annotated to these sex-dependent regions show male-biased enrichment for pathways related to BDNF signalling, in line with male-biased expression changes related to Neuropeptide Signaling". BDNF is not a neuropeptide

Lines 323-325: "Remarkably, motif analysis of these closed regions revealed that nearly 80% of them contain the Egr1 motif (Figure 5h, Suppl.Table 9)". This is not remarkable, this is expected. Knocking down a TF should result in the closure of sites that contain that TF's motif.

Lines 378-382: "While earlier studies emphasized epigenetic regulation of IEGs as part of their activity-dependent regulation, evidence for the role of IEGs as direct epigenetic regulators of their target genes is a newer finding that provides a novel mechanistic insight into IEGs' regulatory role in neuronal adaptation and plasticity." This is simply false. The study of IEG TFs and their target genes goes back decades. AP-1 (Fos/Jun) heterodimers were extensively investigated going back to the 90s and more recently the Greenberg lab and others have comprehensively characterized the targets of Fos, Npas4, and other IEGs in the brain, even with single-cell resolution.

Lines 385-387: "In addition, this is reminiscent of the role of cFos, another IEG, shown in males to direct chromatin opening in a neuronal activity-dependent manner, although whether this occurs in the female brain is not known." Are the authors suggesting that none of the published work on Fos targets was performed in cells from females?

Lines 406-409: "In fact, this cyclical expression in females may also be indicative of Egr1's role in driving the biological clocks in the body more generally, since Egr1 is also induced in the suprachiasmatic nucleus by light stimulation, and has been implicated in circadian rhythm-driven brain plasticity." This connection between oestrus and circadian cycles is speculative.

Lines 425-428: "Considering its sensitivity to sex hormone levels and sex specificity of its actions, Egr1 therefore represents

a master molecular adaptor at the interface of the external and internal environment, with a critical role in sex-dependent neurobiology and psychiatric risk.” The data presented here do not show that changes in Egr1 expression are directly due to sex hormones.

Version 1:

Reviewer comments:

Reviewer #3

(Remarks to the Author)

While I maintain that reports of sex differences should be supported by inclusion of data from males and females in the same statistical analysis, I have no further comments for the authors.

Response to Reviewer's Comments

We would like to express our continued gratitude to the reviewers for their careful review of this manuscript and general enthusiasm for the study. Here, we address all remaining critiques through both changes to the manuscript, including 5 new figure panels (2 main, 3 supplemental) and ~900 words of additional text, as well as in our point-by-point responses below. We feel that the additional data and text included in this revision further strengthen our conclusions and have improved the overall quality of the study.

Reviewer #1 (Remarks to the Author):

Overall, the authors have done an outstanding job of responding to my previous critiques, as well as those of my fellow Reviewers. The new data & analyses provided nicely support the authors' assertions, and the text revisions have greatly improved the clarity of the manuscript, along with providing more precise language (particularly with respect to the structural morphology experiments) so as to not over interpret their findings. I continue to feel that this manuscript is robust and will be of broad interest to the field. As such, I now support publication of this work at Nature Communications.

We thank the reviewer again for their previous suggestions and for their overall support for the revised manuscript.

Reviewer #2 (Remarks to the Author):

The manuscript by Rocks et al describes the effects of overexpressing the transcription factor Erg1 in ventral hippocampus of male and female mice in studies testing the possibility that this IEG regulate chromatin structure and gene expression changes underlying synaptic plasticity and behavioral changes across the female estrous cycle. The original submission received three detailed and extensive reviews that brought up issues of experimental design, the strength of support for some of the conclusions reached and wording. The authors have gone to great lengths to modify the manuscript in response to the review comments including the addition of additional experiments and analyses of publicly available data (response to Reviewer #1), revision of statistical analysis and wording (response to all three reviewers), and changes in interpretation (all three reviewers). The revisions increased support for the conclusions reached and improved language so as to not overinterpret the results. Moreover, specific elements of experimental design were explained and some of the experimental findings were more clearly acknowledged. Overall, it is a strong manuscript, providing novel evidence for consequences of Erg1 overexpression in females which indicate mechanisms through which normal fluctuations in Erg1 expression across the estrous cycle in females may control gene expression.

We would like to thank the reviewer for their enthusiastic support for the revised manuscript and for their previous recommendations which were critical toward improving the original draft.

Regarding response to Reviewer #1 point 3, I agree with the authors in that analysis of overexpression in non-OVX females could give rise to complex and difficult-to-interpret results. In those animals there would be a hugely different hormonal environment that could provide counter balancing influences.

We thank the reviewer for their support on this point. While it is certainly a complex question, it is encouraging that other experts in the field share our view on the appropriate way to analyze and interpret data from this experimental paradigm.

The authors made a number of wording changes, reflecting review comments (e.g., the changes from ‘synaptic plasticity’ to ‘structural plasticity’ and from sex-specific to sex-dependent is important). Also important is the recognition, and now fuller description, that many of the effects of Egr1 overexpression are similar between males and females – this is now noted and more fully considered in the manuscript.

I do think the authors are still selling their results a bit hard at the end of the abstract and discussion but the changes in wording are an improvement.

Line 95: I think “across sex” should be changed to “between the sexes” (also line 116)

We again want to thank the reviewer for their appreciation for the revised manuscript, and we fully agree that the changes in wording and the framing of the results has improved the study significantly. To continue improving the manuscript, we have altered the end of the discussion and the lines mentioned above to address the reviewer’s remaining concerns:

(Lines 464-467):

Since ovarian hormone shifts are an established but underexplored psychiatric risk factor in humans^{3,11,12,56–58}, our study represents an important contribution toward the discovery of sex-based drug targets and treatments for neuropsychiatric disorders.

(Lines 108-109):

While female gonadectomy precluded a direct comparison between the sexes (see Methods), we included age-matched males for a comparison in all experiments.

(Lines 128-131):

Whereas ovariectomy, again, prohibited a direct comparison between the sexes (see Methods), we performed overlaps of Egr1’s transcriptional effects in males with those in OVX females to further isolate female-biased transcriptional mechanisms potentially underlying Egr1’s behavioural effects.

Reviewer #3 (Remarks to the Author):

In their revised manuscript, the authors have added substantial new analyses and an entirely new Egr1 knockdown experiment as suggested by Reviewer 1. I appreciate that they provided the complete lists of their DEGs, GO terms and DARs for their genomic experiments formatted as Supplementary Tables.

We thank the reviewer for their recognition of the efforts dedicated toward revising the original manuscript and the appreciation for the availability of the data that we generated for the reader. We also thank the reviewer for their continued efforts in carefully reviewing our work and articulating their critiques of the study. We want to emphasize that we understand the reviewer’s concerns and have done our best to address the remaining critiques through additional experiments, analyses, and changes to the text, as well as with point-by-point responses below. We feel that these additional changes have further improved the quality of the manuscript.

However, I still have concerns with the overall study design, interpretation and conclusions. As I stated previously, the Egr1 overexpression experiment does not effectively model Egr1 levels in proestrus. Partial rescue of spine and behavior phenotypes with Egr1 overexpression suggest that additional proestrus-dependent mechanisms beyond Egr1 expression contribute to increased spines and decreased anxiety-related behaviors during proestrus. Egr1 overexpression only

captures a fraction of the authors' published proestrus-specific changes in gene expression and chromatin accessibility.

We agree with the reviewer that the proestrus behavioral and vHIP molecular state is complex, and that while our data indicate that Egr1 is an important driver of this state, we agree that other mechanisms are almost certainly involved as well, and may function either independently or synergistically with Egr1. This is why we mention, throughout the text, that Egr1 manipulations elicit partial, rather than total, recapitulations of estrous cycle-dependent states (please see below). As a note, manipulation experiments typically dissect only one component of the system under study so we do not find this concern so worrisome. We always emphasize that the estrous cycle is a complex phenomenon with multiple hormones (not only estradiol!) and underlying mechanisms (not only Egr1) certainly being involved.

(Lines 27-29):

Importantly, Egr1 overexpression and knockdown partially mimic the vHIP chromatin state associated with the high and low-oestrogenic phase of the oestrous cycle, respectively.

(Lines 230-231):

Egr1 overexpression partially recapitulates the proestrus-associated transcriptional program in vHIP neurons

(Lines 252-253):

Egr1 overexpression partially recapitulates proestrus-associated chromatin changes in vHIP neurons

(Line 365):

Egr1 knockdown partially reverses the proestrus chromatin state in females

(Lines 377-380):

Enrichment analysis on these overlapping genes revealed that Egr1 knockdown partially recapitulates a dioestrus-like chromatin state by reversing the Egr1-mediated gained-accessibility of neuronal chromatin around a subset of genes related to synapse regulation, dendrites, and oestrogen response.

(Lines 899-900):

Egr1 overexpression partially recapitulates proestrus-associated changes in gene expression, chromatin, and synaptic plasticity.

(Lines 430-434):

It is also worth noting that, beyond Egr1's cyclical expression, we previously found a similar cyclic gene expression pattern for other epigenetic regulators, including multiple ATP-dependent chromatin remodelling factors, in vHIP neurons across the oestrous cycle¹⁸. Thus, cycling transcriptional regulators appear to be a distinctive feature of the oestrous cycle-driven gene regulation¹⁸ that is likely to establish "the oestrous rhythm" in the cell.

(Lines 427-429):

However, the receptor signalling pathway leading to Egr1 induction remains to be established and, considering the hormonal complexity of the oestrous cycle, multiple hormones and receptors may be regulators of cyclical Egr1 levels.

We also agree that future molecular tools that allow researchers to recreate fluctuations in expression levels that mimic the estrous cycle will be valuable for modeling Egr1 and other important factors with the appropriate temporal dynamics. We already developed cyclical estrogen regimen and these are now new data added to the manuscript (Fig. 1d-f). As for Egr1, we now make this point explicitly in the discussion:

(Lines 437-442):

While the long-lasting viral manipulations of Egr1 levels performed here gave important insights into Egr1's sex-dependent effects on gene regulation and behaviour, the development of new molecular tools that allow for finer temporal control of gene expression will be important for future studies to model the cyclical expression of Egr1 and other epigenetic regulators involved in driving the proestrus state.

Extended Data 9 shows very minimal overlap between DARs that open with Egr1 overexpression and DARs that change across the estrous cycle. Taken together, these results suggest that the Egr1 overexpression experiments are not modeling the physiological increase in Egr1 expression with proestrus.

Extended Data 9 (now Supplementary Figure 9) does not show the overlap between DARs across the two datasets mentioned by the reviewer. To clarify, this figure takes the overlap between Egr1 DARs and estrous cycle DEGs shown in Figure 4d and shows that the majority of these DEGs also undergo chromatin changes across the estrous cycle, indicating that chromatin regulation is an important driver of their expression levels in the context of the estrous cycle. This is now clarified in the text:

(Lines 272-274):

Of these oestrous cycle DEGs that overlap with Egr1 chromatin changes, 68.4% (65/95) also have chromatin changes across the oestrous cycle, indicating the importance of chromatin accessibility in their physiological regulation (Suppl. Fig. 9).

Overexpression will persistently boost Egr1 expression in all neuronal cell types as is apparent in Extended Data Figure 2. The mild effects of Egr1 knockdown experiments in proestrus additionally demonstrate that the overexpression experiments are a bit of a sledgehammer approach.

As with other overexpression methodologies used throughout the neuroscience field, our overexpression of Egr1 expectedly resulted in supraphysiological levels of Egr1. This allowed us to definitively characterize the behavioral and molecular consequences of raised Egr1 levels without the additional variability one would expect from attempting to raise levels to those bordering physiological. Such approaches are commonly used not only in the context of overexpression but also, relevant to this field of study, hormone treatments. For example, in the study by Su et al (Nat. Neurosci, PMID: 28166220), the researchers employed AAVs to overexpress cFos for one week and used this "continuous overexpression" experiment to show that cFos overexpression partially mimics neuronal activity-induced chromatin opening at regions with cFos-binding sites. This is a very similar approach to one we took in our overexpression study. In addition, we and others (e.g. Gegenhuber et al. *Nature* 2022 PMID: 35508660 and Rocks et al Nat Commun, PMID: 35705546) treated mice with 5 µg estradiol benzoate, a dose which has been shown by HPLC/MS to result in supraphysiological levels of serum estradiol (PMID: 35705546). Such manipulations are particularly warranted in the context of ovariectomized female mice, where too low of a dose may be insufficient to overcome lasting adaptations associated with ovariectomy. Our recent strategy of cyclical oestradiol dosing (PMID: 39946826; also used

here as described below) overcomes some of these limitations, though an identical strategy for manipulating expression levels of a specific gene do not yet exist, as mentioned above and now in the text:

(Lines 437-442):

While the long-lasting viral manipulations of Egr1 levels performed here gave important insights into Egr1's sex-dependent effects on gene regulation and behaviour, the development of new molecular tools that allow for finer temporal control of gene expression will be important for future studies to model the cyclical expression of Egr1 and other epigenetic regulators involved in driving the proestrus state.

As the reviewer notes, the knockdown experiments were a subtler manipulation with expectedly more subtle results and therefore serve to complement the overexpression experiments by demonstrating the importance of Egr1's effects in the context of physiologically cycling females. We are very glad that we included this additional experiment that provides critical complementary support to our hypothesis.

I dispute the claim that ovariectomy precludes the inclusion of sex as a variable. Female gonadectomy does not preclude the ability to perform statistical comparisons with males. The authors repeatedly use the term "sex-dependent" and their primary conclusion is that Egr1 acts differently in males and females. Accordingly, the Egr1 overexpression and control comparisons should be made in the males and the ovariectomized females in the same statistical analysis, using a 2 way ANOVA. This should be done throughout the manuscript whenever there are adjacent pairwise comparisons (spines, behavior, genes/DARs). Figure 4 shows that Egr1 overexpression in OVX females boosts spine numbers to similar numbers as in control and OE males. Reviewer 2 makes the same argument in their point 7.

We agree with the general sentiment of the reviewer, as demonstrated by our analysis of the knockdown data which included sex as a factor in the clustering analyses, as well as our prior work which analyzes sex as a factor whenever comparisons are made between intact females and males (e.g. PMID: 37777968). However, we hope that the reviewer will agree and consider for their future work, too, that "sex" is a complex biological variable that consists of multiple sex-related variables, among which gonadal hormone status is likely the most important for shaping sex-biased neurobiology. In the experiment in which we remove gonads from females, we also "strip" sex as a variable from its major component and cannot just use it simply as a "whole" factor in the statistical analysis. In addition, OVX females went through the surgical procedure while males did not, which further complicates "sex" as a variable in this case. After consulting with our statistician colleague, who is an author on the study, we have determined that the analysis described by the reviewer here is not appropriate for the overexpression experiments. We would also like to note that Reviewers 1 and 2 are satisfied with our revisions and have not raised any issues with the analysis in the second round of reviews. In fact, the choice of using OVX animals in the case of overexpression experiment was warranted, as agreed by Reviewer 2 in their comment above. With the multifaceted approach that includes the integration of physiological, pharmacological, and gene manipulation data within the ovarian hormone-depleted and naturally-cycling background we provide robust evidence that the effects of Egr1 are indeed sex-dependent.

Additional points:

In the Egr1 knockdown experiment, the authors claim that genes and DARs that decrease or close with Egr1 knockdown are enriched for estrogen response genes, but the Figures show

minimal enrichment. Figure 5c shows perhaps 5 genes associated with the term “estrogen response early” and Figure 5g indicates between 7.5 and 10 genes located near regions that lose accessibility with Egr1. What are these genes and sites? Are the closing sites near the down-regulated genes? The downregulated genes Sh3bp5 and Rara do not have DARs annotated to them in Supp Table 8a.

We would like to clarify that our claim that “genes and DARs that decrease or close with Egr1 knockdown are enriched for estrogen response genes” is supported by statistical tests, which were significant at $p_{\text{adj}} < 0.05$. In the case of enrichment analysis, these tests assess whether the overlap between a list of genes (e.g. DEGs or genes annotated to DARs) and an annotated gene set (e.g. “Estrogen response early”) exceeds what would be expected by chance. The components of these analyses which contribute to statistical significance are the size of each list, the size of the overlap, and the size of the background. Statistically significant gene enrichment can therefore be observed for even overlaps of relatively small magnitude. We have been very transparent with the way we present our data so there is nothing misleading or inaccurate in our data interpretation, as we can indeed see by reviewer’s ability to predict the number of genes present in enriched terms. However, we agree that the particular genes and sites that are enriched would be of interest to the reader, and we therefore now included them in the Supplementary Table 6c and Supplementary Table 8c. We also note that not all DEGs have an associated DAR, and not all genes annotated to DARs are differentially expressed, as was observed for the overexpression experiment.

Kcnv1 is shown in Figure 6, but there is no DAR associated with this gene in Supp Table 8a.

We thank the reviewer for pointing this out. The DAR associated with Kcnv1 is listed in Suppl. Table 8b, which shows regions identified as having sex-biased chromatin changes based on our clustering analysis. We now clarify in the text where the region associated with Kcnv1 can be found:

(Lines 380-385):

Two illustrative examples are Kcnv1 (Suppl. Table 8b), encoding a voltage-gated potassium channel subunit, and Chd4 (Suppl. Table 8a), encoding a chromatin remodelling enzyme (Fig. 6d). Both genes are associated with a chromatin region that contains an Egr1 binding motif and gains accessibility in OVX females following Egr1 overexpression while losing accessibility in proestrus females following Egr1 knockdown (Fig. 6d).

Extended Data 10 should include controls. The authors should plot signal from the published Egr1 ChIP-seq in all groups. For panel a), this includes the ATAC peaks for GFP control-injected males and females. For panel b), this includes ATAC peaks from the scrambled RNA controls and all phases of the estrous cycle.

This request indicates a potential misinterpretation of the figure which we now clarify both here and in the text. Specifically, Extended Data 10 (Now Supplementary Figure 10) currently shows the Egr1 ChIP-seq signal in the following regions: a) DARs more accessible in the Egr1-overexpression group that contain Egr1 binding sites, b) DARs less accessible in the Egr1 knockdown condition that contain Egr1 binding sites, and c) DARs more accessible in proestrus than diestrus containing Egr1 binding sites. The rationale for this analysis is that in each case, the relevant DARs are enriched for Egr1 binding motifs as determined by motif analysis. We therefore sought to determine whether these motif-containing regions represent regions of actual Egr1 binding determined by ChIP-seq. Such motif enrichment was not observed for the

corresponding control DARs (of which there are none for the knockdown experiment, for example) that the reviewer is requesting. We have now clarified this in the text:

(Line 339-346):

To more definitively link Egr1 genomic binding to its chromatin regulatory functions, we assessed Egr1 binding activity in regions of interest by leveraging previously published Egr1 ChIP-seq data from the male cortex⁴⁰. For this analysis, we focused on regions of chromatin that are putatively regulated by Egr1 based on: i) gained accessibility in the overexpression experiment (Suppl. Fig. 10a); ii) lost accessibility in the knockdown experiment (Suppl. Fig. 10b); and iii) gained accessibility in proestrus compared to dioestrus¹⁸ (Suppl. Fig. 10c). In all cases, we found that these regions are highly enriched for Egr1 binding in vivo (Suppl. Fig. 10).

In the publication for the Egr1 ChIP-seq data used here, it was reported that ~44% of binding sites are at promoters, while here the authors report that “neuronal enhancers are the primary substrate for targeted opening of chromatin by Egr1” in their overexpression experiments. Are the promoter sites not open in the ventral hippocampus, or are they simply not differential between groups?

We thank the reviewer for pointing this out. To answer this question, we overlapped Egr1-bound promoters identified in the cortical ChIP-seq analysis with either all unique ATAC-seq peaks (a, left) or unique Egr1 vs. eGFP DARs (b, right). Below is the Figure that is now included as Suppl.Fig. 11:

We also included the following text:

(Line 350-360):

Since the previous publication on Egr1 binding in the male cortex reported that ~44% of binding sites are at promoters⁴⁰, we wondered whether the observed enhancer enrichment here was due to a general inaccessibility of cortical Egr1-bound promoters in the ventral hippocampus, or whether they are simply not differential between groups. To answer this question, we overlapped Egr1-bound promoters identified in the cortical ChIP-seq analysis with either all unique ATAC-seq peaks (Suppl Fig. 11a) or unique Egr1 vs. eGFP DARs (Suppl. Fig. 11b). This analysis demonstrated that while the majority of cortical promoters bound by Egr1 are accessible in vHIP neurons, very few are differentially accessible between the Egr1 and eGFP conditions. This is consistent with our observation that the gained accessibility surrounding Egr1 motifs that we observed following Egr1 overexpression is targeted to neuronal enhancers, rather than promoters.

The example ATAC-seq library in Supplementary Figure 2 looks under-transposed. Although a nucleosomal ladder is apparent, very little of the library consists of accessible chromatin, defined by Buenrostro et al (2013) as mono- and di-nucleosomes and subnucleosomal fragments. The authors should generate plots of their fragment distribution to ensure that the reads they aligned are predominantly fragments of accessible chromatin, as in Nature Methods papers PMID: 24097267 and PMID: 28846090.

As explained in the Nature Protocols ATAC-seq paper (PMID: 35478247), the presence of higher molecular weight fragments in the electrophoretic trace is not necessarily indicative of under-transposition, and the trace we present in Supplementary Figure 2 (now Supplementary Figure 14) actually resembles the example shown in Fig 3d of that publication which is used to exemplify the clear nucleosomal periodicity representative of a high-quality ATAC-seq library. We have generated the fragment distribution plot for the same library, which is now included as Supplementary Figure 14d. This plot demonstrates this point clearly, as we see that insert sizes predominantly correspond to nucleosome-free, mono-, and di-nucleosomal DNA.

Overstatements throughout the manuscript are distracting. It is not that *Egr1* is “sex-dependent” but that its expression and consequent effects are increased in proestrus. It is likely that many IEGs show sex differences in expression due to sex differences in sex hormones and neural activity. *Fos* was first reported to show sex differences in the rat brain 20 years ago. PMID: 15842237

We agree with the reviewer that the difference in *Egr1* levels (and in particular fluctuating levels!) is one source of sex difference. It is worth mentioning, however, that we previously found numerous examples of proestrus-specific accessibility of *Egr1* binding sites despite similar *Egr1* expression levels in proestrus females and males (PMID: 31253786), which was the first time someone reported IEG-driven sex differences in chromatin accessibility in the literature. In addition, we demonstrate that, for example, an identical manipulation of *Egr1* expression in males and ovariectomized females has differing effects on behavior, gene expression, and chromatin organization. We think it is appropriate to refer to these as “sex-dependent” effects of *Egr1*. We mention that *Egr1*’s effects for certain outcomes are not female-exclusive in the discussion:

(Line 451-453):

These findings explain why we see extensive chromatin and gene expression changes in both sexes with partial overlaps in molecular and structural plasticity, indicating sex-dependent but not female-exclusive effects by Egr1.

Additional overstatements include the following:

Line 52. “So far, molecular studies in neuroscience have focused on the male brain. Please omit this sentence. Many if not most modern molecular neuroscience studies include females, in line with SABV requirements from major funding institutions, and there has been extensive work over the past decade on mechanisms of estrogen signaling in the female rodent brain. The cited reference 14 is from 2011.

While we agree that some progress has been made since the cited reference was published, this is still a pervasive problem in neuroscience and we would argue that “many” modern neuroscience studies including females still falls short of the goals of the SABV initiative. It is also worth mentioning that while SABV requires researchers to mention inclusion of females in their grants, there is no mechanism requiring researchers to actually follow through on this, and studies including males only are still published everywhere including in major journals. Take a recent neuroepigenetics paper in Science (PMID: 39052786) as an example. Lastly, the consequences of females being historically excluded from neuroscience studies has lasting effects that can’t be discounted simply because more recent studies include females, considering the corpus of knowledge is still primarily derived from studies focusing on the male brain. To clarify that this statement concerns primarily the latter point, we have altered it as follows and included three more recent articles that are addressing the issue of female exclusion from the neuroscience literature.

(Line 52-53):

Historically, molecular studies in neuroscience have focused on the male brain^{2,12,14,15}, and our knowledge of sex-specific molecular mechanisms in the brain is therefore limited.

Lines 182-184: “Pathway analysis on genes annotated to these sex-dependent regions show male-biased enrichment for pathways related to BDNF signalling, in line with male-biased expression changes related to Neuropeptide Signaling”. BDNF is not a neuropeptide

This is a semantic point and while peptides are usually smaller in size than BDNF, experts in the field typically still refer to BDNF as a neuropeptide considering it is a gene product that transmits signals across synapses. Consider, for example, the first line of the BDNF signaling review article published in *Cell* by Wang, Kavalali, and Monteggia (PMID: 34963057):

“Brain-derived neurotrophic factor (BDNF) is a neuropeptide that plays numerous important roles in synaptic development and plasticity.”

Lines 323-325: “Remarkably, motif analysis of these closed regions revealed that nearly 80% of them contain the Egr1 motif (Figure 5h, Suppl.Table 9)”. This is not remarkable, this is expected. Knocking down a TF should result in the closure of sites that contain that TF’s motif.

We disagree with the statement that “Knocking down a TF should result in the closure of sites that contain that TF’s motif”. First, we still know very little about pioneering factors, particularly in neuroscience. After reviewing the literature, we have concluded that this has only been demonstrated for select pioneering factors in drosophila (e.g. Zelda: PMIDs 26335634) and cell culture (e.g. Sox2: PMID 37691488). The knockdown of an immediate early gene transcription factor in neurons of the mammalian brain leading to chromatin closure surrounding its motif was only shown in the manuscript of Su et al which we already cite (PMID: 28166220). They showed that knockdown of cFos attenuated neuronal activity induced-chromatin opening at cFos-binding

sites. We hope the reviewer can recognize the novelty and the difficulty of our approach in which we performed the knockdown of an IEG in the naturally-cycling hormonal background in which the expression of the studied IEG varies. This is a difficult experiment where we needed to track the estrous cycle phase following surgery and sacrifice the animals in the right phase of the cycle (proestrus). Another compensatory mechanisms could have easily interfered with our results and it is remarkable to be able to show closure in chromatin after a 50% knock-down of a TF under physiological conditions in neurons of the live brain. In fact, Su et al induced neuronal activation via electroconvulsive stimulation and while this is an outstanding experiment and result, our approach was the only viable option within the specific demands of our study, which has not previously been explored in this context.

Lines 378-382: “While earlier studies emphasized epigenetic regulation of IEGs as part of their activity-dependent regulation, evidence for the role of IEGs as direct epigenetic regulators of their target genes is a newer finding that provides a novel mechanistic insight into IEGs’ regulatory role in neuronal adaptation and plasticity.” This is simply false. The study of IEG TFs and their target genes goes back decades. AP-1 (Fos/Jun) heterodimers were extensively investigated going back to the 90s and more recently the Greenberg lab and others have comprehensively characterized the targets of Fos, Npas4, and other IEGs in the brain, even with single-cell resolution.

The examples given by the reviewer refer to the target genes of TFs encoded by IEGs being identified. We agree that there has been quite a lot of work on this in the field already. However, as quoted by the reviewer, we are referring specifically to “the role of IEGs as direct epigenetic regulators of their target genes”. To our knowledge, the only other studies to show something similar are Su et al. Nat Neurosci 2017 (PMID: 28166220) which shows that Fos overexpression in males increases accessibility around Fos binding sites, and Sharma et al. Neuron 2019 (PMID: 30846309), now cited in our manuscript, which shows that cKO of Npas4 prevents activity-dependent H3K27ac at Npas4 binding sites. If there are additional examples that we are unaware of, we would be happy to include them as references in our manuscript.

Lines 385-387: “In addition, this is reminiscent of the role of cFos, another IEG, shown in males to direct chromatin opening in a neuronal activity-dependent manner, although whether this occurs in the female brain is not known.” Are the authors suggesting that none of the published work on Fos targets was performed in cells from females?

We thank the reviewer for pointing this out, we mistakenly left the appropriate citation from this sentence, which is the above-mentioned study by Su et al. (PMID: 28166220), which was performed on dentate gyrus tissue from male mice. We have now included this reference in the text:

(Line 411-413):

In addition, this is reminiscent of the role of cFos, another IEG, shown in males to direct chromatin opening in a neuronal activity-dependent manner¹⁵, although whether this occurs in the female brain is not known.

As stated above, other than this study by Su et al., none of the published work on Fos targets that we are aware of describes the ability of Fos to “direct chromatin opening”, and are therefore not relevant to the sentence.

Lines 406-409: “In fact, this cyclical expression in females may also be indicative of Egr1’s role in

driving the biological clocks in the body more generally, since *Egr1* is also induced in the suprachiasmatic nucleus by light stimulation, and has been implicated in circadian rhythm-driven brain plasticity.” This connection between oestrus and circadian cycles is speculative.

We agree that this is a speculative sentence. However, the word “may” in the sentence makes it clear that this is an intriguing possibility, rather than a factual claim. *Egr1* has been implicated, though, in circadian rhythm-driven brain plasticity, so our discussion is well-placed.

Lines 425-428: “Considering its sensitivity to sex hormone levels and sex specificity of its actions, *Egr1* therefore represents a master molecular adaptor at the interface of the external and internal environment, with a critical role in sex-dependent neurobiology and psychiatric risk.” The data presented here do not show that changes in *Egr1* expression are directly due to sex hormones.

We thank the reviewer for this important point. While we have shown that *Egr1* expression levels fluctuate over the estrous cycle, we did not directly implicate any specific sex hormones in cycle-dependent regulation. To address this, we performed an additional experiment in which proestrus females were compared to 2 groups of ovariectomized females: OVX-EB received 1ug estradiol benzoate (EB) every 4 days for 6 weeks to model cyclical estradiol replacement, and OVX-Veh received vehicle injections along the same schedule. We found that this cyclical treatment restores proestrus-like levels of *Egr1* in the ventral hippocampus, directly linking estradiol levels to the regulation of vHIP *Egr1* expression. Further, cyclical EB treatment partially rescued OVX-induced deficits in the time spent in the center of the open field, which is consistent with the partial behavioral rescue observed with *Egr1* overexpression in OVX females. The newly generated data can be found in Figure 1d-f and the results are described in the text:

(Line 88-98):

We further wanted to provide evidence that oestradiol levels directly regulate vHIP Egr1 expression and behaviour across the oestrous cycle (Fig. 1d-f). To this end, we used “cyclical” treatment with oestradiol benzoate (EB; 1µg s.c. every 4 days) in ovarian hormone-depleted, ovariectomized (OVX) mice for 6 weeks (Fig. 1d) mimicking the physiological rise in oestradiol occurring every 4 days in cycling females²⁰ (Fig. 1a). While removal of ovaries in young adult mice (at 8 weeks) led to reduced vHIP Egr1 mRNA levels (Fig. 1e) and increased anxiety-related behaviour in the open field (Fig. 1f) compared to age-matched, high estrogenic (proestrus) cycling mice, the cyclical oestrogen replacement restored proestrus-like levels of Egr1 expression (Fig. 1e, Suppl. Table 1) and partially rescued anxiety-related behaviour (Fig. 1f, Suppl. Table 1) in OVX mice. These results confirm the role of cyclical oestradiol in regulating vHIP Egr1 levels and behaviour.

We would like to again thank the reviewer for this point which prompted us to generate additional data which has further strengthened the conclusions of the manuscript and our overall understanding of how hormones drive changes in chromatin and gene expression. We hope that the reviewer will appreciate all the effort put into this manuscript, its importance to the field, and the integration of physiological, pharmacological, and gene manipulation data within the ovarian hormone-depleted and naturally-cycling background which is setting an example for our field, And we again thank all the reviewers for helping us produce such comprehensive and strong manuscript.

Response to Reviewer's Comments

We thank Reviewer 3 for their support and acceptance of our previous revision, which aimed to address all of their remaining concerns. Below, we address the one outstanding concern from the reviewer.

Reviewer #3 (Remarks to the Author):

While I maintain that reports of sex differences should be supported by inclusion of data from males and females in the same statistical analysis, I have no further comments for the authors.

Thank you for making this point. We maintain our position that the covariate “sex” in a statistical test is misleading when one sex is gonadectomized, and that the knockdown experiment which was performed in intact females and does include a group-by-sex analysis addresses this concern. However, upon reviewing the manuscript we noted that this limitation of the overexpression experiment and corresponding strength of the knockdown experiment was not clearly addressed in our discussion. We have therefore extended the discussion to address this point more clearly.

Lines 417-431:

“While ovariectomy allowed us to isolate the effect of ovarian hormones in females, it also “stripped” sex as a variable of its major component and precluded using sex as a factor in our statistical analysis of the overexpression experiments. However, the knockdown experiment provided a complementary approach in cycling females, allowing for analysis across sex in intact male and female mice. With the multifaceted approach that includes the integration of physiological, pharmacological, and gene manipulation data within the ovarian hormone-depleted (overexpression) and naturally-cycling (knockdown) background we provide robust evidence that the effects of Egr1 are indeed sex-dependent.

In fact, our results show that the cyclicity of ovarian hormone levels may provide a critical, sex-specific signal for Egr1's effect on chromatin regulation in females. Egr1's response to ovarian hormones seems to be optimized by oestrogen's regulation of Egr1 expression, as reflected in the cyclical Egr1 expression pattern across the oestrous cycle in the vHIP, but also in other brain regions such as the prefrontal cortex⁴⁷ or peripheral tissues such as uterus⁴⁸, as well as in the restoration of proestrus-like vHIP Egr1 levels observed when we treated OVX females with cyclical oestradiol regimen.”